# Direct RNA sequencing reveals m$^6$A modifications on adenovirus RNA are necessary for efficient splicing

Alexander M. Price [1], Katharina E. Hayer [2], Alexa B. R. McIntyre[3,4,13], Nandan S. Gokhale[5,14], Jonathan S. Abebe[6], Ashley N. Della Fera[1,15], Christopher E. Mason [3,7,8,9], Stacy M. Horner [5,10], Angus C. Wilson[11], Daniel P. Depledge[6,16✉] & Matthew D. Weitzman [1,12,16✉]

Adenovirus is a nuclear replicating DNA virus reliant on host RNA processing machinery. Processing and metabolism of cellular RNAs can be regulated by METTL3, which catalyzes the addition of *N6*-methyladenosine (m$^6$A) to mRNAs. While m$^6$A-modified adenoviral RNAs have been previously detected, the location and function of this mark within the infectious cycle is unknown. Since the complex adenovirus transcriptome includes overlapping spliced units that would impede accurate m$^6$A mapping using short-read sequencing, here we profile m$^6$A within the adenovirus transcriptome using a combination of meRIP-seq and direct RNA long-read sequencing to yield both nucleotide and transcript-resolved m$^6$A detection. Although both early and late viral transcripts contain m$^6$A, depletion of m$^6$A writer METTL3 specifically impacts viral late transcripts by reducing their splicing efficiency. These data showcase a new technique for m$^6$A discovery within individual transcripts at nucleotide resolution, and highlight the role of m$^6$A in regulating splicing of a viral pathogen.

[1] Division of Protective Immunity and Department of Pathology and Laboratory Medicine, The Children's Hospital of Philadelphia, Philadelphia, PA 19104, USA. [2] Department of Biomedical and Health Informatics, The Children's Hospital of Philadelphia, Philadelphia, PA 19104, USA. [3] Department of Physiology and Biophysics, Weill Cornell Medicine, New York, NY 10065, USA. [4] Tri-Institutional Program in Computational Biology and Medicine, New York, NY 10065, USA. [5] Department of Molecular Genetics and Microbiology, Duke University Medical Center, Durham, NC 27710, USA. [6] Department of Medicine, New York University School of Medicine, New York, NY 10017, USA. [7] The HRH Prince Alwaleed Bin Talal Abdulaziz Alsaud Institute for Computational Biomedicine, Weill Cornell Medicine, New York, NY 10065, USA. [8] The World Quant Initiative for Quantitative Prediction, Weill Cornell Medicine, New York, NY 10065, USA. [9] The Feil Family Brain and Mind Research Institute, Weill Cornell Medicine, New York, NY 10065, USA. [10] Department of Medicine, Duke University Medical Center, Durham, NC 27710, USA. [11] Department of Microbiology, New York University School of Medicine, New York, NY 10017, USA. [12] Department of Pathology and Laboratory Medicine, University of Pennsylvania Perelman School of Medicine, Philadelphia, PA 19104, USA. [13]Present address: Department of Molecular Life Sciences, University of Zurich, 8006 Zurich, Switzerland. [14]Present address: Department of Immunology, University of Washington, Seattle, WA 98115, USA. [15]Present address: Biological Sciences Graduate Group, University of Maryland, College Park, MD 20742, USA. [16]These authors jointly supervised this work: Daniel P. Depledge, Matthew D. Weitzman. ✉email: daniel.depledge@nyulangone.org; weitzmanm@email.chop.edu

Adenovirus is a nuclear-replicating DNA virus with a linear double-stranded genome that is dependent on the host cell machinery for productive infection[1]. To maximize coding capacity of the 36 kilobase genome, adenovirus employs a tightly regulated gene transcription process. Early genes and subsequent late genes are produced from both strands of DNA using cellular RNA polymerase II and the spliceosomal machinery to generate capped, spliced, and polyadenylated messenger RNAs (mRNA). Besides the four canonical ribose nucleosides, adenoviral RNA is also known to contain RNA modifications[2,3]. RNA modifications comprise a family of over one hundred chemical modifications of nucleic acid that can play important roles in both RNA biogenesis and function[4,5]. In eukaryotic messenger RNA, $N6$-methyladenosine ($m^6A$) is the most prevalent RNA modification besides the 7-methylguanosine cap[6]. This mark has been implicated in regulating multiple processes of RNA maturation, including splicing, polyadenylation, export, translation, and ultimately decay[7–12]. Current understanding suggests that $m^6A$ is added to messenger RNAs co-transcriptionally in the nucleus by recruitment of a writer complex composed of METTL3, METTL14, WTAP, and other accessory proteins to RNA polymerase II[13–15]. These modified mRNAs are bound by reader proteins such as the YTH family (YTHDC1-2, YTHDF1-3)[16,17], various hnRNPs[18,19], and the IGF2B[20] family of RNA binding proteins, which affect various downstream fates of the modified mRNAs. This mark is reversible through the action of erasers, and FTO and ALKBH5 have been proposed to demethylate $m^6A$[8,21,22].

Shortly after the discovery of $m^6A$ in cellular RNAs, RNAs from several diverse viruses such as adenovirus, Rous sarcoma virus, simian virus 40, herpes simplex virus, and influenza A virus were also shown to contain $m^6A$ in internal regions[2,3,23–26]. In particular, adenovirus serotype 2 was shown to contain $m^6A$ at sites away from the cap[2]. These marks were added to nuclear pre-mRNA and retained in the fully processed cytoplasmic RNA[3]. These two studies also predicted adenovirus RNAs to contain on average four m6A modifications per transcript, along with low levels of methyl-5-cytosine, for a total of 1/400 (0.25%) of modified viral nucleotides[2,3]. However, no subsequent studies have elucidated the functions of $m^6A$ modification in adenovirus. With the advent of high-throughput $m^6A$ sequencing methods, interest in viral RNA modifications has been rekindled. RNA viruses such as HIV, influenza A, picornavirus, and various *Flaviviridae* including Zika, dengue, and hepatitis C virus are influenced both positively and negatively by $m^6A$ added via METTL3, and many of these viral RNAs are bound by cytoplasmic YTHDF proteins[27–34]. In hepatitis B virus, $m^6A$ at the same site can both stimulate reverse transcription, as well as reduce mRNA stability[35]. For DNA viruses such as SV40 and KSHV, deposition of $m^6A$ on viral RNA transcripts can enhance viral replication[36–39]. Interestingly, multiple labs have published conflicting functions for $m^6A$ within the same viral transcript of KSHV, which suggests cell type specific roles[39]. Of note, recent work using human cytomegalovirus also implicates $m^6A$ in controlling aspects of the interferon response, thereby indirectly regulating viral infection[40,41]. Since adenovirus is reliant on cellular polymerases and mechanisms to generate and process its viral RNAs, adenovirus infection provides an excellent opportunity to study the consequences of co-transcriptional $m^6A$ addition.

Until recently, sequencing methods to map $m^6A$ have relied on antibody-based immunoprecipitations to enrich for methylated RNA within a relatively large nucleotide window (methylated RNA immunoprecipitation sequencing, meRIP-seq or $m^6A$-seq)[42,43]. These techniques are indirect, because antibody-precipitated RNA has to be converted to cDNA before sequencing. Although other RNA modifications can be located

due to mutations or truncations resulting from reverse transcription[44–46], these events are not generated in the case of $m^6A$ due to efficient base pairing with thymine and uracil. Several techniques have circumvented some of these limitations, such as photo-crosslinking assisted $m^6A$ sequencing (PA-m$^6$A-Seq)[47], $m^6A$ individual nucleotide resolution crosslinking and immuno-precipitation (miCLIP)[48,49], and RNA digestion via $m^6A$ sensitive RNase (MAZTER-seq)[50]. In general, these methods are labor intensive, and require either specialized chemical addition to cell culture, large amounts of input material, or higher unique read counts than meRIP-seq. Furthermore, the antibodies used to precipitate $m^6A$ may themselves have sequence or structure biases, and cannot distinguish between $m^6A$ and the similar modifications $m^6A_m$[22,51]. To this end, the ability to sequence native RNA molecules directly using nanopore arrays provides a new approach to locate RNA modifications. While detecting modified DNA nucleotides is possible using both PacBio and Oxford Nanopore Technologies platforms[52,53], detection of RNA modifications has proven much more challenging. Recently, two groups have shown detection of $m^6A$ using nanopores in yeast total RNA and in human cell lines[54,55]. In addition to detecting RNA modifications directly, production of long reads by these platforms provides distinct advantages in the study of gene-dense viral genomes, which encode complex and often overlapping sets of transcripts[56]. To date, the ability to use direct RNA sequencing to map full-length transcripts and their RNA modifications unambiguously has not been realized.

In this study, we found that adenovirus infection does not alter expression of $m^6A$-interacting enzymes but instead concentrates these host proteins at sites of nascent viral RNA synthesis. While meRIP-Seq was able to identify numerous methylated regions on both early and late kinetic classes of viral mRNA, the complex splicing structure and overlapping nature of the adenovirus transcriptome precluded unambiguous transcript assignments and $m^6A$ localization by this method alone. To overcome this limitation, we developed a method to predict sites of $m^6A$ modification at single-base resolution within full-length RNA by direct RNA sequencing and used this technique to predict $m^6A$ specific to transcript isoforms. While we found that both viral early and late genes are marked by $m^6A$, expression of viral late RNAs in particular decreased dramatically with loss of the cellular $m^6A$ writer METTL3. This late gene-biased effect was primarily mediated by decreased RNA splicing efficiency in the absence of METTL3, and could be extended to all of the multiply spliced adenovirus late RNAs. Overall, these results highlight a new technological advancement in long-read RNA sequencing, and reveal that $m^6A$ influences the splicing and expression from a viral pathogen.

## Results

**Nuclear $m^6A$-interacting factors concentrate at viral RNA during adenovirus infection.** While it is known that adenovirus RNA transcripts contain $m^6A$, the impact of infection on cellular components involved in writing, reading, or erasing $m^6A$ is unknown. The majority of $m^6A$ on messenger RNA is installed co-transcriptionally in the nucleus by a writer complex composed minimally of METTL3, METTL14, and WTAP that associates directly with RNA Pol II[13–15]. To examine whether $m^6A$-interacting enzymes are altered during adenovirus infection, we performed immunoblot analysis over a time-course of infection with adenovirus serotype 5 (Ad5) in A549 lung adenocarcinoma cells (Fig. 1a). Over the course of infection, levels of the assayed writers (METTL3, METTL14, and WTAP) and readers (YTHDC1, YTHDF1, and YTHDF2) remain unchanged. There was a modest increase in levels of both purported erasers, FTO

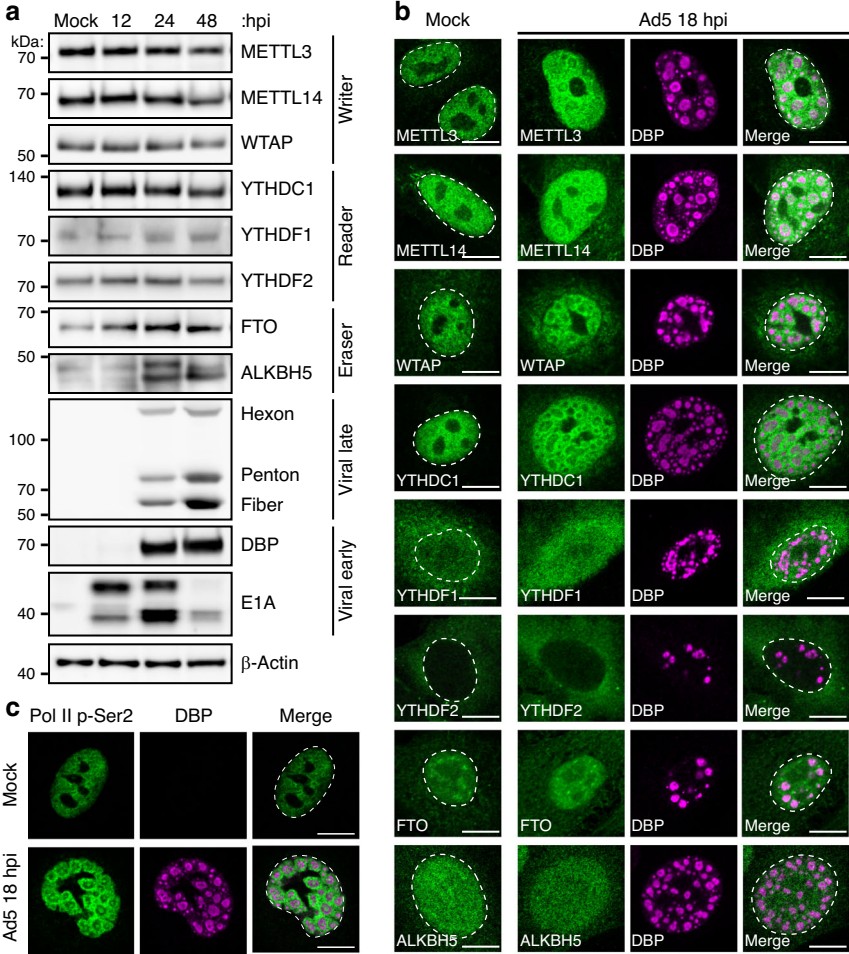

**Fig. 1 Nuclear m6A-interacting factors concentrate at sites of nascent viral RNA synthesis. a** Abundance of cellular m6A proteins is unchanged during infection. Immunoblot showing abundance of m6A writers, readers, and putative erasers over a time-course of adenovirus infection. Viral early (E1A and DBP) and late (Hexon, Penton, and Fiber) proteins demonstrate representative kinetic classes. β-Actin is the loading control. Kilodalton size markers shown on the left. **b** Confocal microscopy of m6A-interacting proteins (green) in mock-infected or Ad5-infected A549 cells 18 h post-infection (hpi). DBP (magenta) is the viral DNA binding protein that marks sites of nuclear viral replication centers. The nuclear periphery is shown by a dotted white line as assessed by DAPI staining. Scale bar = 10 μm. **c** Confocal microscopy showing the pattern of actively transcribing RNA Polymerase II phosphorylated on serine 2 of CTD (Pol II p-Ser2, green) in mock-infected cells or relative to DBP (magenta) in infected cells. Scale bar = 10 μm. All data are representative of at least three independent experiments.

and ALKBH5, including the appearance of a faster migrating band detected with the ALKBH5 antibody.

Adenovirus is known to recruit specific cellular factors to viral replication centers or mislocalize antiviral cellular factors[57,58]. To determine if localization of m6A-interacting factors were similarly altered, we performed indirect immunofluorescence microscopy to localize cellular proteins, as well including an antibody against the viral DNA binding protein (DBP) to demarcate viral replication centers (Fig. 1b). When comparing mock-infected A549 cells to cells infected with Ad5 for 18 h, we observed that METTL3, METTL14, WTAP, and YTHDC1 relocalized from their diffuse nuclear pattern into ring-like structures surrounding the sites of viral DNA replication marked by DBP. These structures have been previously characterized as sites of viral RNA transcription[59], and were consistent with staining for phospho-serine 2 on the RNA Pol II C-terminal domain, a marker of actively transcribing polymerase (Fig. 1c). The localization of cytoplasmic readers (YTHDF1, YTHDF2) and demethylases (ALKBH5 and FTO) was mostly unchanged. These data highlight that while adenovirus does not significantly alter expression levels of known m6A writing enzymes, these nuclear

proteins are concentrated at sites of viral RNA synthesis, and not actively excluded or mislocalized, during infection.

**Adenovirus transcripts contain METTL3-dependent m6A modifications.** While it is known that adenovirus mRNAs contain m6A[2,3], it is not known exactly where these marks are located or whether adenovirus infection affects m6A localization within host transcripts. To address these questions, we performed meRIP-seq on poly(A)-selected RNA from A549 cells that were mock-infected or infected with Ad5 for 24 h (Supplementary Data 1). Strand-specific sequencing was performed on fragmented RNA immunoprecipitated (IP) with an anti-m6A antibody, as well as on total input RNA. We used the MACS2 algorithm to call peaks in IP over input reads for both viral and cellular transcriptomes across three biological replicates (Fig. 2a and Supplementary Data 2). HOMER motif analysis revealed the characteristic DRACH signature (Where D = A/G/U, R = A/G, and H = A/U/C) as the highest ranked motif in cellular m6A peaks in both mock and infected conditions, indicating that the immunoprecipitation was successful and that use of the canonical

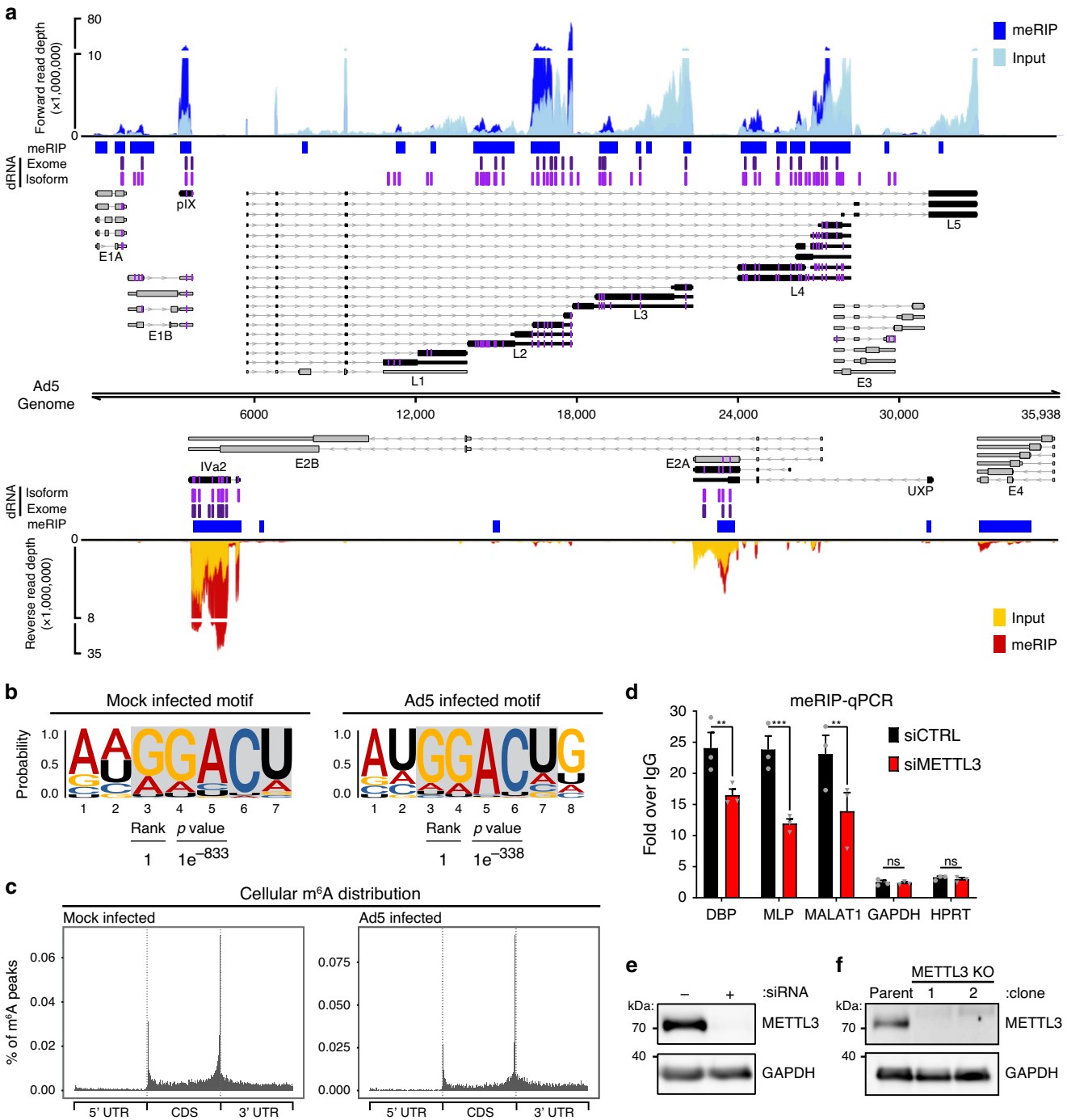

**Fig. 2** (caption not shown)

m6A motif was unperturbed by infection (Fig. 2b). Furthermore, the general location of m6A peaks in cellular transcripts was unchanged during infection and showed the characteristic stop codon/3'UTR bias in mRNA metagene plots (Fig. 2c). We next focused on m6A addition to viral transcripts and identified 19 peaks in the forward transcripts and 6 in the reverse transcripts (Fig. 2a). While these peaks covered every viral kinetic class and transcriptional unit produced by the virus, the MACS2-generated peak areas were very broad, and it was impossible to identify which of many overlapping viral transcripts were m6A methylated. Multiple peak callers were used, but ultimately none were successful in deconvoluting the large peaks due to the nature of overlapping short reads from multiple distinct viral transcripts. MACS2 peaks were retained for downstream comparison due to

the robust peak calling ability of this software as tested using other meRIP-seq datasets[60].

We next asked whether viral RNAs were m6A-methylated by cellular enzyme METTL3. To achieve this, METTL3 was knocked down by siRNA prior to Ad5 infection of A549 cells and meRIP followed by qRT-PCR of total RNA was performed at 24 hs post-infection (Fig. 2d, e). These results were normalized for the amount of input RNA, and demonstrate that the amount of m6A-marked RNA available for immunoprecipitation were all reduced in a METTL3-dependent manner for the early viral transcript DBP and late viral RNAs generated from the Major Late Promoter (MLP), as well as the positive control cellular transcript *MALAT1*. These data indicate that viral early and late transcriptional units contain m6A, and that this modification is added by

**Fig. 2 Transcript-specific analysis reveals adenovirus RNAs contain METTL3-dependent m⁶A modifications. a** The viral transcriptome is schematized with forward facing transcripts above the genome and reverse transcripts below. Viral gene kinetic classes are color-coded to denote early (gray) or late (black) genes. Lines with arrows denote introns, thin bars are untranslated exonic regions, and thick bars represent open reading frames. The names of each viral transcriptional unit are shown below the transcript cluster. meRIP-Seq was performed in triplicate on Ad5-infected A549 cells at 24 hpi. Representative meRIP data (blue/red) and total input RNA (light blue/yellow) sequence coverage is plotted against the adenovirus genome. Peaks containing increased meRIP-seq signal over input were called with MACS2 and denoted by blue boxes. Using direct RNA (dRNA) sequencing, full-length RNAs were sequenced from A549 parental cells or METTL3 knockout cells infected with adenovirus for 24 h. Specific m⁶A sites were predicted by comparing the nucleotide error rate of dRNA sequence data from WT to KO cells. Indicated in purple vertical lines are individual adenosines predicted to be modified by m⁶A that reach statistical significance when applied to all RNA that maps to a single nucleotide of the Ad genome (dRNA Exome). All Ad5-mapping transcripts were binned into unique full-length reads spanning entire transcript isoform and the same m⁶A prediction was applied on a transcript-by-transcript basis. Magenta vertical lines indicate predicted m⁶A residues found on the transcriptome level (dRNA Isoform). In addition, the position of m⁶A present in each viral transcript is highlighted in magenta directly on the transcript schemes. **b** HOMER reveals nucleotide motifs through analysis of MACS2 called peaks in cellular meRIP-seq data from Mock or Ad5-infected samples. Statistical significance was determined using hypergeometric enrichment calculations to find enriched motifs, and p-value was corrected for multiple testing. **c** Metagene analysis of m⁶A-peak distribution across cellular mRNA molecules containing 5′ and 3′ untranslated regions (UTR) and coding sequence (CDS) in Mock or Ad5-infected samples. **d** meRIP-qRT-PCR was performed on total RNA isolated from Ad5-infected control or METTL3 knockdown A549 cells 24 h post-infection. **e** Immunoblot showing knockdown efficiency of METTL3 in A549 cells. **f** Representative immunoblot showing two clones generated from Cas9-mediated knockout of METTL3 in A549 cells. For all assays, significance was determined by unpaired two-tailed Student's T-test, **$p \leq 0.01$, ***$p \leq 0.001$, ns = not significant. Exact p-Values are included in the source data file. Sequencing experiments are representative of three biological replicates for Illumina data and two biological replicates analyzed in a four-way comparison for Nanopore data. Immunoblots in panels **e** and **f** were independently performed at least three times. Graphs represent mean +/− standard deviation.

the cellular writer complex that includes METTL3. However, the exact location of the m⁶A mark could not be assessed by this approach.

**A statistical framework predicts sites of m⁶A methylation using direct RNA sequencing.** To address shortcomings of short read-based m⁶A-sequencing platforms, we sought a technique that would provide both single nucleotide resolution, as well as long read length to allow for unambiguous assignment of m⁶A sites to specific adenovirus mature transcripts. Nanopore sequencing has been used to call DNA modifications directly using differences between measured and expected current values as nucleotides travel through the pore[53]. It is well established that signal deviations during direct RNA sequencing can result from the presence of one or more base modifications, and that this leads to an increase in base-call error rate around the modified base[54,61]. The likelihood of a given nucleotide being modified (i.e., carrying an m⁶A mark) can be assessed by a $2 \times 5$ contingency table to examine the distribution of base-calls between two datasets (e.g., m⁶A positive and m⁶A negative) at a given genome position, as was recently demonstrated[62]. Here, a G-test on the distribution of A, C, G, U, and indels provides a score and p-value that requires subsequent (Bonferroni) correction for multiple testing (Fig. S1a). This approach is codified in the software package DRUMMER (https://github.com/DepledgeLab/DRUMMER).

To produce m⁶A positive and negative datasets, we generated METTL3 knockout A549 cells using CRISPR-Cas9 with a strategy that included regulated expression of a nuclease-insensitive transgene (Fig. 2f). We performed direct RNA sequencing using two biological replicates each of RNA collected from parental wild-type A549 cells (WT) and METTL3 knockout A549 cells (M3KO), each infected with Ad5 for 24 h (Fig. 3a and Supplementary Data 3). Since the datasets are unlinked, each METTL3 KO dataset was compared to each WT parental dataset, yielding four distinct comparisons (Fig. S1b). Each comparison yielded between 335 and 452 candidate sites with significant G-test statistics (Supplementary Data 4). To account for only the positions at which the base error rate was greater in the WT (m⁶A positive) dataset, we filtered for a one-fold or greater increase in the ratio of mismatch:match base-calls compared to the M3KO (m⁶A negative) dataset. This reduced the number of putative sites to 191–217 (Fig. S1b). Reasoning that multiple sites proximal to a

single m⁶A modification could show significant differences (adjusted $p < 0.01$ in error rate), we next calculated the distance between each candidate site and its nearest neighbor candidate in a strand-specific manner (Fig. S1c). We determined that the majority of candidate sites had at least one neighboring m⁶A candidate site within five nucleotides, so we implemented an additional filtering strategy to collapse all clustered candidates to one candidate site (Fig. 3b), retaining only the candidate site with the highest G-test statistic. We subsequently plotted the distance (number of nucleotides) from the identified base to the nearest upstream or downstream AC motif (the minimal possible m⁶A motif). Most significant candidate sites filtered for mismatch:match rates (94.2–99%) were located within five nucleotides of an AC motif (Fig. 3c), the maximum distance at which a modified base is thought to affect basecall error[61]. Masking reduced this fraction slightly (92.7–98.9%, Fig. 3c) and yielded 89–111 distinct predicted m⁶A sites, of which 53 were conserved across all four dataset comparisons (Fig. 3d, highlighted by dark blue line in Fig. 2a). The majority (83.1%) of these 53 sites mapped directly to adenines in AC motifs, with the remainder mapping within four nucleotides (Fig. 3c). When random non-candidate nucleotides were selected from our dataset we found that the distance to the nearest AC motif was much greater than for m⁶A-candidate sites, indicating that this was not due to random chance (Fig. S1d). We subsequently extracted the seven base sequence centered on a collapsed candidate m⁶A site and generated a sequence logo (Fig. S1e) that closely matched the m⁶A DRACH logo that we confirmed for m⁶A modifications in the human transcriptome. Since all m⁶A candidate sites mapped to within five nucleotides of nearby AC, subsequent shifting of the motif center to the closest AC revealed that only 13% of sites did not perfectly recapitulate the DRACH motif (Fig. 3e). Of these non-canonical motifs, three were DRACG and the remaining four were GUACU. The overlap between antibody-based meRIP predicted regions and our dRNA-based approach was significantly higher than expected by chance (Fig. S1f-g). Concordance between the predicted location of these m⁶A sites and the previously established motif supports the validity of this unbiased mapping approach.

**Exome versus isoform-level m⁶A analysis.** After identifying putative m⁶A-modified bases within the viral RNA exome using direct RNA-seq, we extended our approach to transcript isoform-

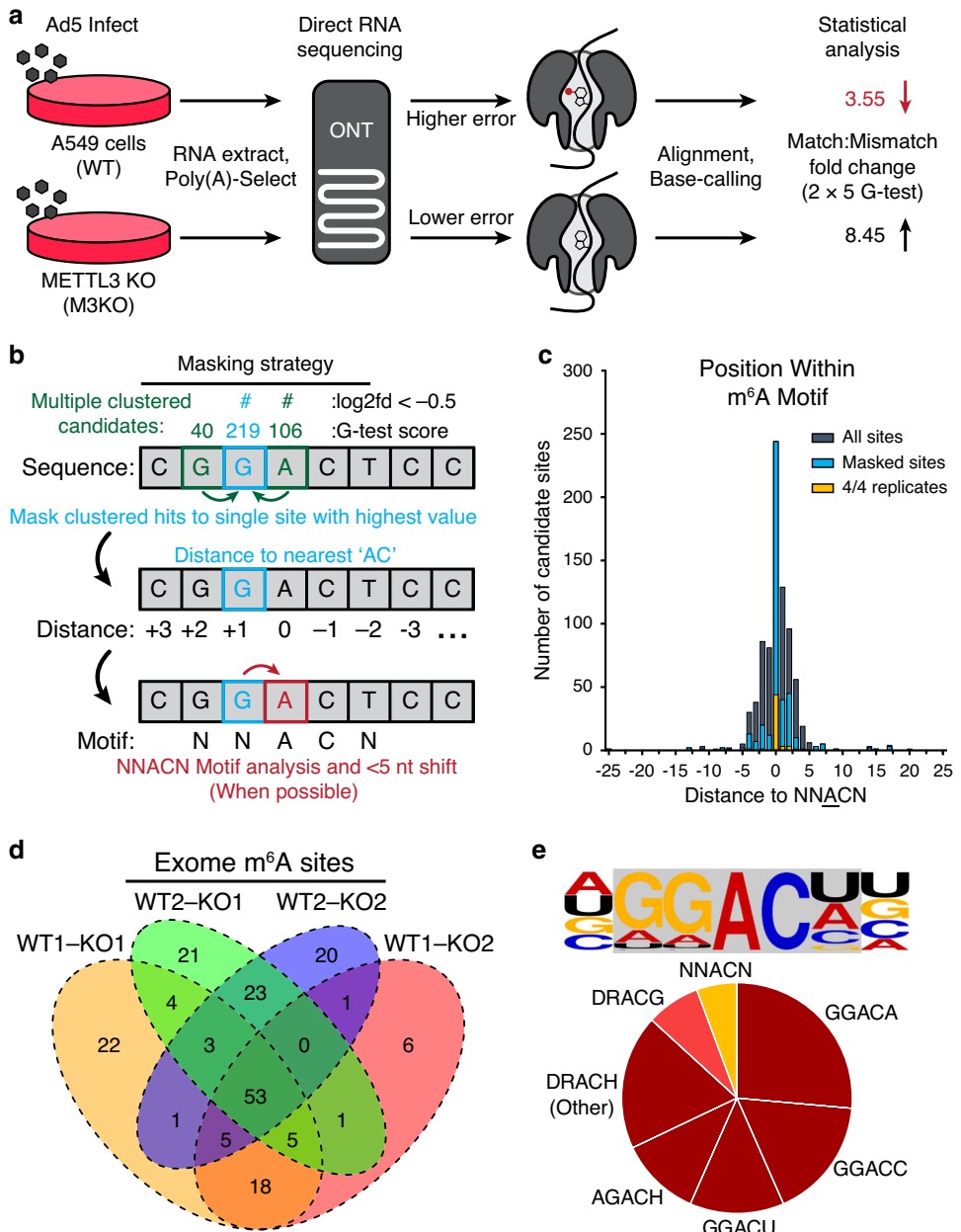

**Fig. 3 A statistical framework for m6A detection using direct RNA sequencing. a** Schematic diagram of proposed strategy to detect m6A sites using direct RNA sequencing. RNAs generated in WT cells (METTL3 positive) will contain m6A modifications, while RNA from METTL3 KO cells (M3KO) will not. As modified RNA passes through the pore there will be a higher error rate during base-calling compared to unmodified RNA. When compared to the reference transcriptome, the aggregate fold change in the Match:Mismatch ratio will be lower in at nucleotides containing m6A. **b** Proposed strategy for masking neighboring candidates. Here, three sites within five nucleotides of an AC produce significant G-test scores. All candidates are collapsed to the single candidate within five nucleotides giving the highest G-test statistic. Collapsed/masked candidates are analyzed for their distance to nearest 'AC' dinucleotide. When nearest 'AC' dinucleotide is within the five-nucleotide window (dictated by nanopore size) the candidate is shifted to the closest 'A' within an 'AC' core, if possible. **c** For each significant candidate site with a one-fold or greater difference in the match:mismatch ratio, the distance to the nearest AC motif was calculated and plotted (gray). This was repeated after masking neighboring candidates (blue) and for the 53 genome-level sites identified across all four comparisons (gold). **d** Comparisons between WT and M3KO (KO) datasets yields 184 putative m6A modified bases (post-collapse) of which 53 are consistently detected across all four comparisons. **e** The consensus five-nucleotide motif for putative m6A modified bases in the Ad5 transcriptome is predominantly comprised of four common DRACH motifs.

level analysis. Here, we aligned our nanopore sequencing reads against 75 distinct transcripts derived from the recently re-annotated adenovirus Ad5 genome[63] and observed that the aligner (MiniMap2[64]) produced secondary alignments for many reads and supplementary alignments for a smaller subset of reads (Fig. S2a). Secondary alignments indicate that a region of a given sequence read aligns with high confidence against two or more

distinct transcripts while supplementary alignments indicate potentially chimeric reads where two segments of the same read align to separate overlapping transcripts (Fig. 4a). Given the possibility that overlapping transcript isoforms may share the same DRACN motifs but still undergo differential methylation, we reasoned that reads with multiple alignments could reduce sensitivity of detection. We thus retained only reads that

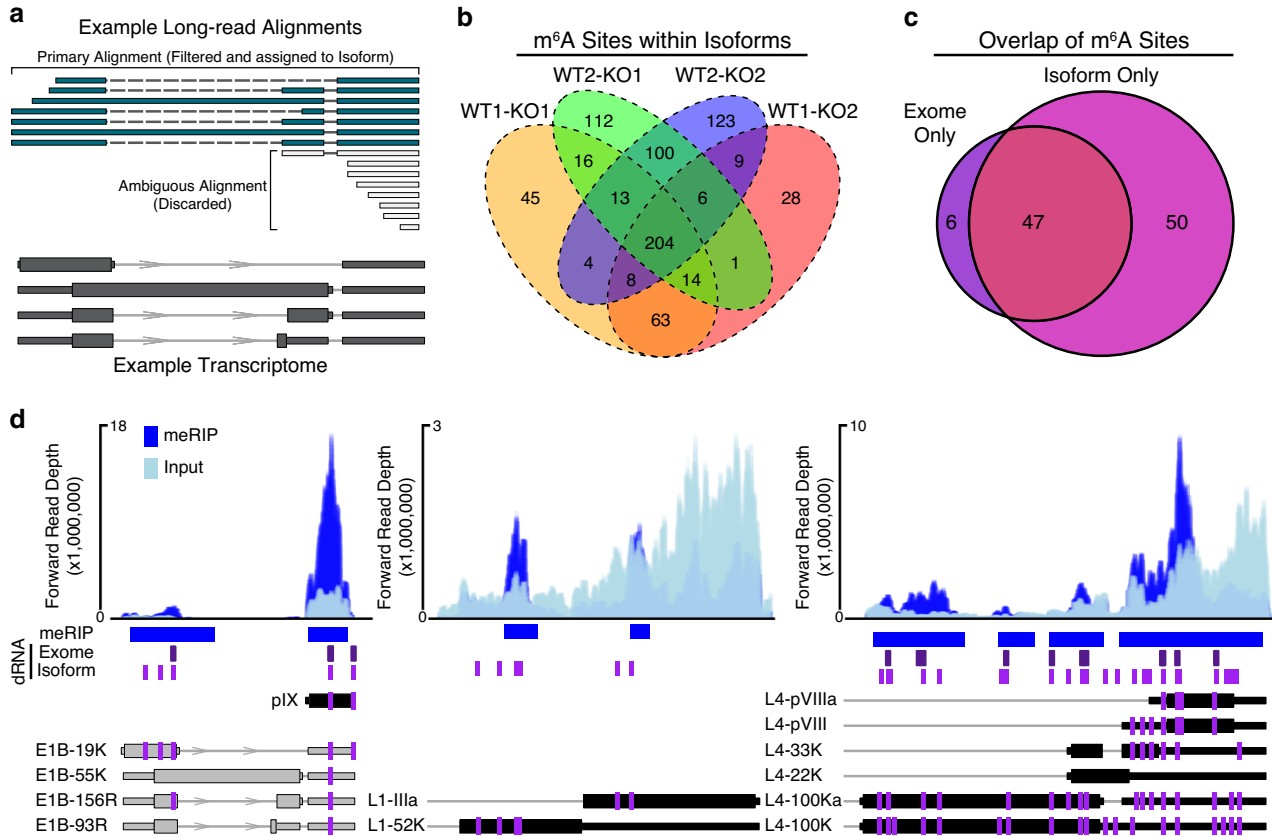

**Fig. 4 Exome versus isoform-level m⁶A analysis. a** Example of read filtering. Overlapping RNA isoforms are shown in dark gray. Sequence reads that map unambiguously (teal) are filtered and retained for m⁶A analysis. Sequence reads with multiple primary alignments (gray) are discarded if no single alignment is considered superior. **b** Isoform-level comparisons between WT and M3KO datasets yields 747 candidates (post-collapse) of which 204 are consistently detected across all four comparisons. **c** Overlap of all m⁶A sites detected in 4/4 exome-mapped replicates and 4/4 isoform-mapped replicates. Isoform only candidates were collapsed to unique genomic loci. **d** Representative plots showing isoform-specific m⁶A predictions on adenovirus transcripts. MACS2 peaks and meRIP-seq signal (blue) is shown on top. Direct RNA (dRNA) predicted m⁶A sites at the exome-level (purple) or isoform-level (magenta) are shown below meRIP peaks as vertical lines. The positioning of m⁶A within individual viral transcripts from Fig. 2 are shown as magenta lines. Transcript names are shown to the left of each transcript.

produced unique alignments with mapping qualities >0 and had no insertions greater than 20 nucleotides, with the latter requirement intended to exclude reads from incompletely spliced RNAs. While this approach discarded 63–69% of our sequence reads (Fig. S2a), the specificity achieved by unambiguous isoform-level assignment of sequence reads allowed us to identify 747 putative m⁶A-modified bases across 47 transcripts. This translated to 352 exome-level sites, of which 204 were conserved across all pairwise comparisons (Fig. 4b and Supplementary Data 5, highlighted by magenta lines in Fig. 2a). Isoform-level analysis recapitulated the majority of exome-level sites identified above (47/53), while identifying an additional 50 m⁶A sites (Fig. 4c). This represents a four-fold increase when compared to the equivalent exome-level analysis, demonstrating the greater sensitivity of the isoform-level approach. Furthermore, this strategy allowed us to detect transcripts that had unique m⁶A sites, even compared to overlapping transcripts that shared the same potential DRACH motifs (Fig. 4d). To validate our data further, we also identified putative m⁶A sites using NanoCompore which performs comparative interrogation of signal level direct RNA-Seq data (dwell time and current intensity) to predict modified nucleotides[65]. Using the same datasets as inputs, we identified 204 putative m⁶A sites at the isoform level: 93 of these were also reported by DRUMMER in all four comparisons, while a further 69 were also reported by DRUMMER in 1-3 comparisons. This left just 41 sites predicted by NanoCompore that were not

identified by DRUMMER compared to 42 sites predicted by DRUMMER but not NanoCompore (Fig. S2b, c). Overall, these data highlight a novel technique that reveals m⁶A marks at both single nucleotide and isoform-specific levels, greatly improving our ability to map m⁶A modifications across complex viral transcriptomes.

**Loss of METTL3 and m⁶A methylation differentially impacts late viral gene expression.** Since both viral early and late genes are marked by METTL3-dependent m⁶A, we asked what role this mark might play in the viral infectious cycle. METTL3 or METTL14 were knocked down by siRNA 48 h prior to infection, a time point that led to stable loss of both RNA and protein expression throughout the subsequent infection (Fig. S3a, b). As reported by others, siRNA-mediated knockdown of METTL3 results in a concomitant loss of METTL14 protein without affecting mRNA, and vice versa[14] (Fig. 5a, Fig. S3b). Upon knockdown of either METTL3 or METTL14, we observed reductions of viral late proteins Hexon, Penton, and Fiber. In contrast, the viral early protein DBP was largely unaffected (Fig. 5a), even though the DBP transcript is also marked by m⁶A (Fig. 2a). When we assessed viral genome amplification by qPCR, we detected only a minimal change after knockdown of METTL3 or METTL14 (Fig. 5b), indicating m⁶A is not required for early stage adenovirus infection. This contrasts with robust decreases in

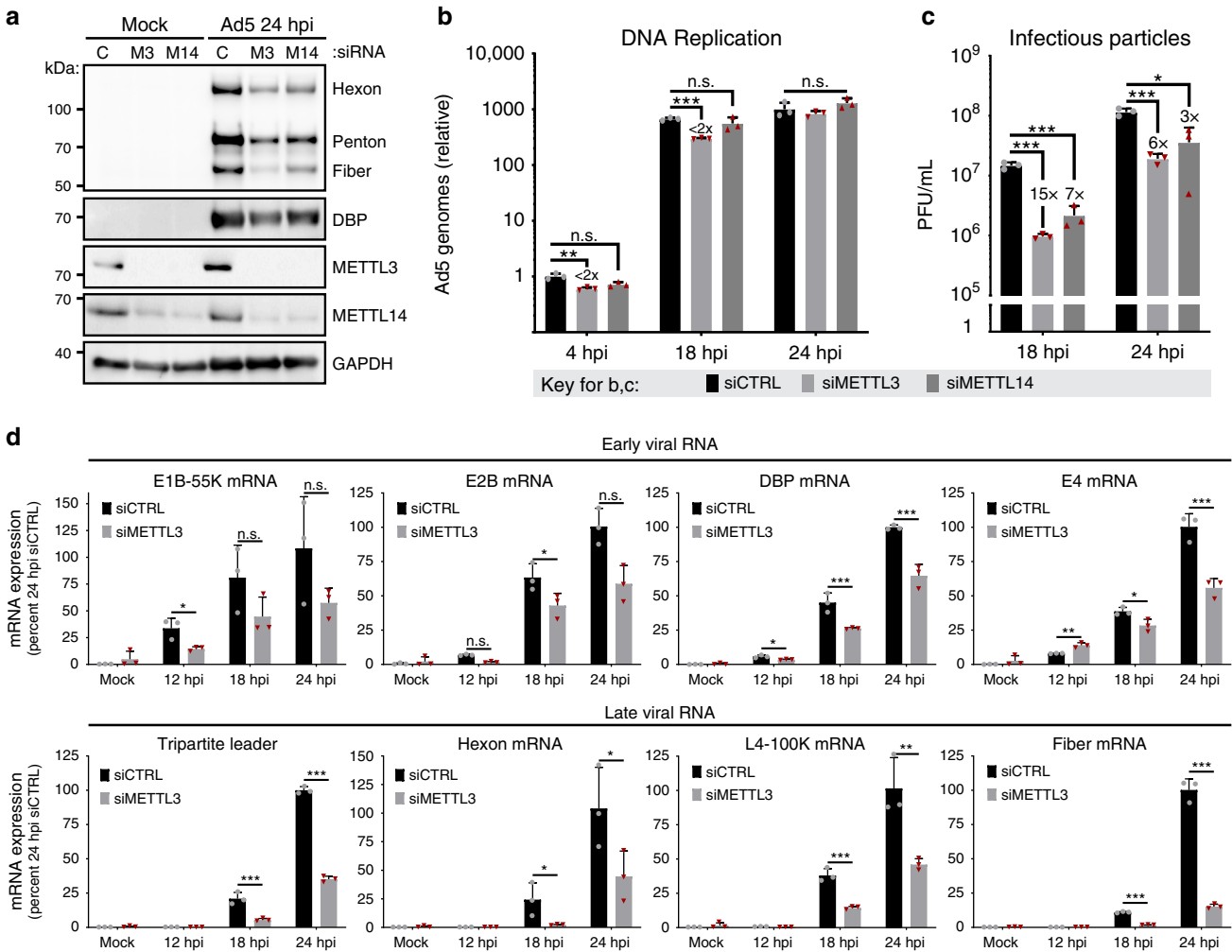

**Fig. 5 Loss of METTL3 differentially impacts late viral gene expression. a** A549 cells were transfected with control siRNA (C), or siRNA targeting METTL3 (M3) or METTL14 (M14) for 48 h before Ad5 infection. Viral late proteins were analyzed at 24 hpi with polyclonal antibody against structural proteins (Hexon, Penton, and Fiber) or early protein DBP. **b** Viral DNA replication was measured in biological triplicate samples with qPCR at different times after infecting A549 cells depleted of METTL3 or METTL14 by siRNA. Fold-change in genome copy was normalized to the amount of input DNA at 4 hpi in control siRNA treated cells. **c** Infectious particle production was measured at different times by plaque assay after infecting A549 cells depleted of METTL3 or METTL14 by siRNA. **d** Viral RNA expression was measured by qRT-PCR in a time-course of infection after control or METTL3 knockdown for 48 h. Viral early transcripts are shown on the top, while late stage transcripts are shown on the bottom. For all assays significance was determined by unpaired two-tailed Student's $T$-test, $*p \leq 0.05$, $**p \leq 0.01$, $***p \leq 0.001$, ns = not significant. Exact $p$-values are included in the source data file. Experiments are representative of three biological replicates and graphs represent mean +/− standard deviation.

the number of infectious particles in lysates derived from infected cells knocked down for either METTL3 or METTL14, as measured by plaque assay (Fig. 5c). Consistent with these findings, when we used reverse transcription coupled with quantitative PCR (qRT-PCR) to measure total viral mRNA levels over a time course of infection (Fig. 5d), we observed that early viral genes were only modestly (less than two-fold or not significantly) decreased by METTL3 knockdown, whereas viral late genes were significantly reduced (sometimes greater than 10-fold). To corroborate these findings from siRNA-mediated knockdown, we also assayed specific stages of the viral infectious cycle in METTL3 KO A549 cells. Two independently generated CRISPR-mediated knockout cells were infected with Ad5 for 24 h and assayed for viral early and late genes by immunoblotting and qRT-PCR. Consistent with siRNA findings, accumulation of DBP transcript and protein was not affected, but viral late gene products were significantly reduced in METTL3 KO cells (Fig. S3c, d). Finally, we blocked methylation by treating cells with the small molecule inhibitor 3-Deazadenosine (DAA) at the time of

infection. DAA is an S-Adenosylmethionine synthesis inhibitor that has preferential effects on $m^6A$ deposition when used at low concentrations[66]. Although treatment with DAA reduced early gene protein expression and DNA replication more than METTL3 knockdown or knockout, the most robust effects observed were reductions in the amount of viral late gene transcripts, protein production, and plaque forming units (Fig. S3e-h). Overall, these data highlight a preferential effect of $m^6A$ addition at late stages of the viral infectious cycle, with viral early gene transcription and genome replication largely unaffected while late RNAs, late proteins, and infectious progeny production are greatly reduced in the absence of $m^6A$.

Two groups have recently highlighted that in primary cells $m^6A$ destabilizes the *IFNB1* transcript, such that loss of METTL3 results in increased production of IFNβ[40,41]. When these cells were infected with viruses, including by adenovirus serotype 4, they restricted viral infectivity through indirect priming of the interferon pathway. To determine if loss of METTL3 similarly increased *IFNB1* and innate immune-related transcripts in our

experiments, we used siRNA to knock down METTL3 for 48 h prior to infection with Ad5 either in the presence or absence of ruxolitinub, a JAK/STAT inhibitor that blocks signaling downstream of type-I interferons such as IFNβ (Fig. S4a). As a positive control for interferon activation and the efficacy of ruxolitinub, uninfected cells were transfected with poly(I:C) as a surrogate viral RNA agonist. Our data demonstrate that in A549 cells, neither type-I interferon (*IFNB1*) or type-III interferon (*IFNL1*) transcripts are induced by Ad5 infection or METTL3 depletion, whereas poly(I:C) transfection induces these transcripts several hundred-fold. Furthermore, expression of interferon-stimulated genes such as *MX1* and *OAS1* was blocked downstream of poly(I:C) transfection by JAK inhibition but not induced by viral infection or METTL3 knockdown. In addition, blocking the interferon pathway with JAK inhibition did not alter the specific decrease in viral late gene we report here (Fig. S4b). Overall these data suggest a specific role for m6A in altering accumulation of late adenoviral RNA transcripts at late stages of the adenoviral infectious cycle, independent of innate immune activation, in the cell types we used.

**Cytoplasmic readers and m6A erasers do not affect the adenoviral infectious cycle.** Since we found that loss of the m6A writer METTL3 resulted in decreases in viral late gene transcripts and proteins, we next examined whether loss of m6A erasers led to increases in viral protein production. We used siRNA to knockdown either ALKBH5 or FTO in A549 cells and performed a time-course infection with Ad5. There was no defect in either viral early proteins or late proteins with either ALKBH5 or FTO knockdown (Fig. S5a). Furthermore, when assaying viral DNA replication by qPCR or infectious particle production by plaque forming units after knockdown of the two erasers, we saw less than 3-fold or non-significant changes (Fig. S5b, c). Using siRNA in A549 cells, we were able to knock down the cytoplasmic m6A reader proteins YTHDF1-3 efficiently (Fig. S5d) but did not observe changes in viral protein production (Fig. S5e). These results highlight that whatever amount of m6A deposition is necessary for adenoviral function, removal of ALKBH5 and FTO does not increase m6A beyond necessary levels.

**Splicing efficiency of late viral RNA is mediated by m6A.** We investigated various mechanisms to explain the differential impact of m6A on early and late viral RNAs. To rule out the role of transcription in masking effects of this post-transcriptional modification, we examined early and late promoter activity by profiling nascent RNA transcription with a 4-thiouridine (4sU) labeling approach[67]. After 24 h infection of A549 cells with Ad5 following knockdown of METTL3, nascent RNA was labeled with 4sU for the last 10 min of infection. Total RNA was harvested, nascent RNA conjugated with biotin, and precipitated for analysis by qRT-PCR with primers to viral transcripts. We examined unspliced regions of viral RNA near the early E1A and E4 regions, as well as within the tripartite leader of the Major Late Promoter (MLP) that precedes nearly all adenovirus late genes. While relative transcription of the E1A promoter was increased two-fold after METTL3 knockdown, a separate early promoter (E4), and the viral late promoter itself showed no significant change (Fig. 6a). RNA half-life was assessed by performing transcriptional shut-off assays with the RNA Pol II elongation inhibitor 5,6-Dichloro-1-β-d-ribofuranosylbenzimidazole (DRB) to assess the stability of viral RNAs. We quantified RNA abundance over an 8 h timecourse in the presence or absence of METTL3, and observed no significant change in RNA decay (Fig. 6b). These data imply that the same amount of late stage viral mRNA are transcribed in adenoviral infections of cells lacking METTL3, and

that despite similar spliced RNA stability fewer late viral RNAs still accumulate.

Besides transcription from the MLP, which is dependent on viral DNA replication, the other major difference between early and late viral transcripts is the sheer number of alternative splicing and polyadenylation events that must take place to generate mature mRNAs. While most early transcripts contain only one splice junction and no more than two potential poly(A) sites, the major late transcriptional unit has over 25 splice acceptors and 5 major sites of polyadenylation, leading to over 20 functional open reading frames[63]. Mounting evidence suggests that m6A and the nuclear reader YTHDC1 can regulate mRNA splicing[7,37]. We used qRT-PCR to measure spliced and unspliced viral RNAs and determined the splicing efficiency of a particular transcript as a ratio of these two products (Fig. 6c, Fig. S6a-c). We examined the m6A-marked transcripts E1A (early gene) and Fiber (late gene) as representative of their kinetic classes. Of note, Fiber is the only late gene that can be incorporated into this type of qPCR assay, since no other intron amplifying primer set within the late gene transcriptional unit is specific to a single gene. Upon knockdown of METTL3 or WTAP, the splicing efficiency of the E1A gene did not change, whereas the splicing efficiency of Fiber significantly decreased (Fig. 6d). METTL3-dependent loss of late gene splicing efficiency was independent of time post infection (Fig. S6e, f). The effect on late gene splicing, but not early gene splicing, was phenocopied by METTL3 knockout (Fig. 6e, Fig. S6d). Fiber RNA splicing efficiency decrease was also seen by knockdown of the nuclear m6A reader protein YTHDC1 (Fig. 6f). This decreased splicing efficiency appears to explain the total RNA decrease in viral late genes, as well as the decrease in protein expression, since both METTL3 and WTAP knockdown lead to significant losses in viral late gene proteins (Fig. 6g). A similar result was observed with knockdown of YTHDC1, although to a much lesser degree (Fig. S6g, h).

To assay the role of m6A in *cis* on viral transcript splicing we turned to a transgene splicing assay in which we can ablate m6A sites (Fig. 6h). In this system, we generated a construct with an exogenous plasmid-based promoter to drive expression of an RNA containing the splice donor exon of the viral late tripartite leader, as much intervening Fiber intron as possible without including additional viral genes or splice sites, and the 5' region of Fiber that encompasses the meRIP-seq peak. This viral cassette was fused to a *Renilla* luciferase gene in which all m6A DRACH motifs were silently mutated[34]. These constructs allow transgene expression with wild-type viral context (WT Fiber), or with all 15 DRACH sites present in both exonic and intronic regions silently mutated to ablate deposition of METTL3-dependent m6A (m6A Mut Fiber, mutations shown below in Fig. 6h). Potential intronic sites were also ablated, since both meRIP-seq and direct RNA sequencing were performed on polyadenylated mRNA and would not have detected potential m6A sites within introns. HeLa cells were depleted of METTL3 with siRNA for 48 h before the respective transgene plasmids were transfected into uninfected cells and incubated for a further 24 h. Using this system, splicing efficiency is read out with qRT-PCR using the primers for endogenous Fiber splicing. Upon knockdown of METTL3, the WT Fiber construct showed a decrease in splicing efficiency (Fig. 6i), similar to that observed during viral infection (Fig. 6d). Importantly, m6A Mut Fiber showed a similar decrease in splicing efficiency when compared to the WT Fiber construct during control siRNA knockdown, and had no further decrease upon METTL3 knockdown (Fig. 6i). The same transgene splicing assay was performed following YTHDC1 knockdown (Fig. S6i). A similar decrease in splicing efficiency of the wildtype reporter was observed, with no change in the m6A mutated reporter. There was no change in the accumulation of an unspliced Firefly

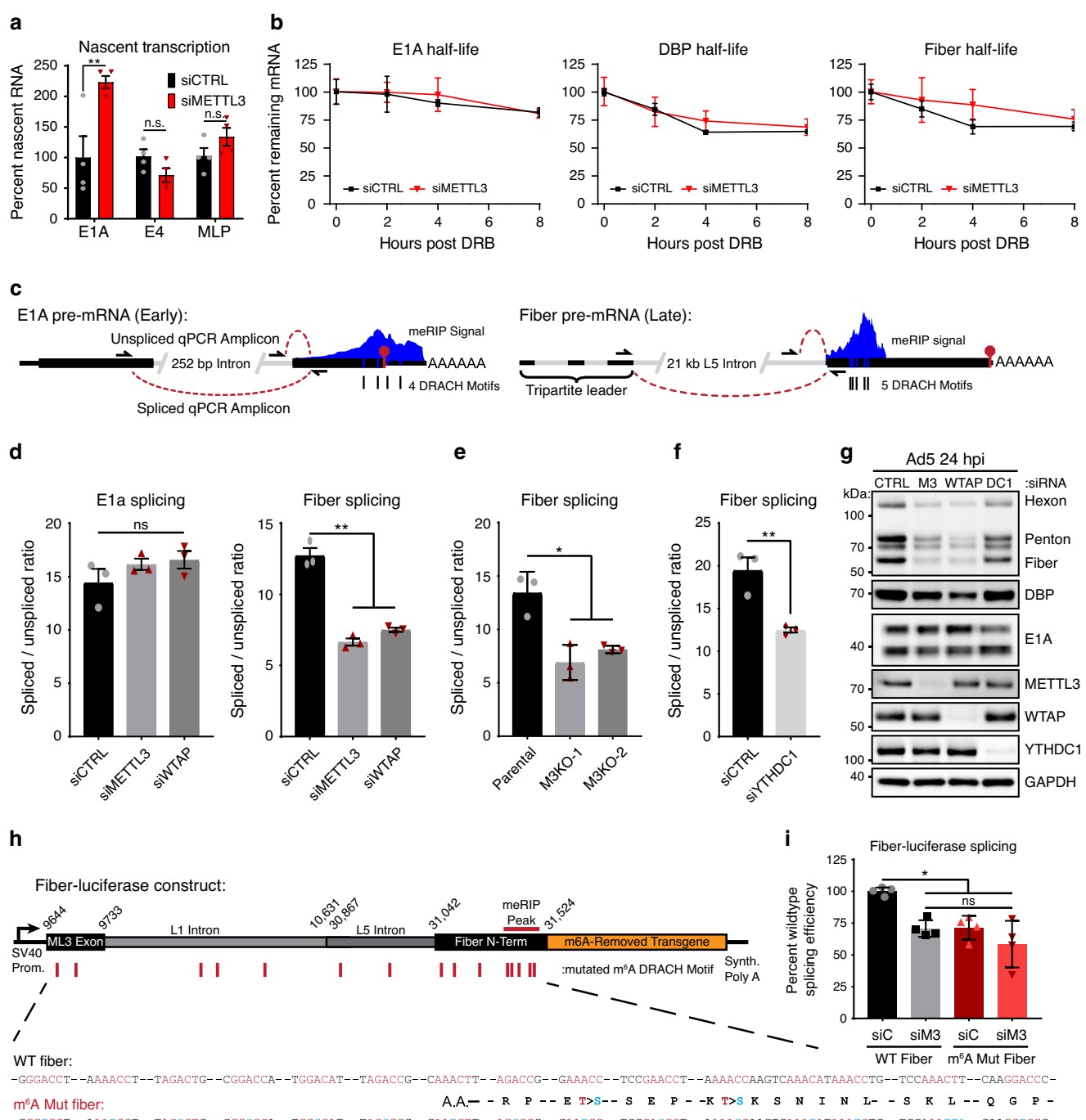

luciferase RNA expressed from the same plasmid upon METTL3 knockdown (Fig. S6j). Overall, these data suggest m6A positively regulates the splicing reaction of viral late transcripts.

**METTL3 knockdown globally dysregulates adenoviral late RNA processing.** To determine if m6A positively regulates the splicing of all late transcripts, we returned to two orthologous sequencing technologies, short-read sequencing and long-read direct RNA sequencing. Due to the limitations of short-read mapping to viral genomes with overlapping transcripts, standard expression algorithms such as mapped fragments per kilobase transcript per million base pairs (FPKM) cannot be used to quantify these viral transcripts accurately. Instead, we focused exclusively on the short reads that contained splice junctions,

allowing unambiguous assignment to one viral transcript. Since our RNA-seq library preparation was strand-specific, this allowed us to map reads accurately to their strand of origin. Furthermore, we were able to address intron retention of the entire late transcriptional unit by counting only unspliced reads overlapping the major splice donor exiting the conserved third exon of the tripartite leader (intron retention, IR). This strategy is schematized for top-strand viral genes (Fig. 7a). When we applied this technique to adenovirus-infected A549 cells after METTL3 knockdown, we observed that the prevalence of early gene splice junctions was largely unchanged (Fig. 7b). Of these, E1B-19K appears to be an outlier, since the prevalence of this particular splice junction increases as the virus moves into the late stage, and a potential ORF of this transcript encodes for protein IX, an alternate promoter-driven adenovirus late gene. When we

**Fig. 6 Splicing efficiency of late viral RNA is mediated by m6A. a** Nascent transcription was analyzed by labeling RNA with 1 mM 4-thiouridine (4sU) for exactly 10 min at 24 hpi for infections of A549 cells transfected with control (siCTRL) or siRNA-mediated knockdown of METTL3 (siMETTL3). Nascent 4sU-labeled RNA was extracted for use in qRT-PCR for analysis of relative transcription rates of two viral early genes (E1A and E4) and the tripartite leader (MLP) found in all Ad5 late transcripts. Samples include four biological replicates. **b** Transcriptional shut-off was performed by adding 60 μM of the RNA Pol II elongation inhibitor DRB to the media of cells infected with Ad5 for 24 h, and stability of labeled spliced transcripts was measured by qRT-PCR for 2, 4, or 8 h post shut-off. Samples include three biological replicates. **c** Schematic of the early transcript E1A and the late transcript Fiber that both contain m6A sites. Three primers allow for the distinction between spliced and unspliced PCR products that can be analyzed by qRT-PCR. **d** Splice efficiency as defined by the relative ratio of spliced to unspliced transcripts of E1A and Fiber were analyzed by qRT-PCR in A549 cells infected with Ad5 for 24 h after depletion of METTL3 or WTAP. Data represents three biological experiments. **e** Fiber splice efficiency was analyzed in Parental A549 cells or two independent METTL3 KO cell lines. **f** Splice efficiency of Fiber was analyzed after siRNA-mediated depletion of the nuclear m6A reader YTHDC1. **g** Immunoblot showing viral late proteins Hexon, Penton, and Fiber, as well as viral early proteins DBP and E1A. A549 cells were depleted of METTL3 (M3), WTAP, YTHDC1 (DC1) or control siRNA (siC) for 48 h prior to infection with Ad5 for 24 h. Immunoblot representative of three independent experiments. **h** Schematic design for a luciferase construct that expresses the third adenovirus tripartite leader to Fiber splice site with intervening L1 and L5 adenoviral intron. The 5′ fragment of Fiber that contains the m6A signal peak was fused in-frame to a *Renilla* luciferase transgene where all m6A DRACH motifs have been ablated by silent mutation. A matching construct was generated with all 15 potential m6A DRACH motifs ablated by silent or synonymous mutation (m6A Mut Fiber). **i** HeLa cells were control transfected (siC) or depleted of METTL3 (siM3) by siRNA for 48 h before transfection with either WT Fiber or m6A Mut Fiber plasmid. At 24 h after the second transfection, splicing efficiency of the transgene was assayed using Fiber-specific primers. Data represent four biological replicates. For all assays significance was determined by unpaired two-tailed Student's *T*-test, *$p \leq 0.05$, **$p \leq 0.01$, ns=not significant. Exact p-Values are included in the source data file. Graphs represent mean +/− standard deviation.

performed this analysis on late gene splice sites, we observed that every late gene originating from the major late promoter was significantly decreased, and that intron retention in this transcriptional unit was increased, consistent with our previous results. The only exception we detected was the first possible alternative splice junction encoding L1-52K, which was unchanged. While the increase in intron retention was not as dramatic as the decrease in spliced late transcripts, this can be explained by the relative instability of unspliced RNA transcripts within the nucleus[68], and the bias of the poly(A) selection employed for sequencing in capturing mature transcripts.

While short reads targeting splice junctions provided high depth quantitative results, we turned to long-read sequencing with unambiguous transcript mapping as an orthogonal technique to validate these results. We counted full-length RNA transcripts after infection of METTL3 knockdown A549 cells normalized to the read depth of the experiment (Fig. 7c). Confirming the short-read sequencing data, we saw that early transcripts were essentially unaffected by loss of METTL3, while the overall abundance of every viral late transcript, except L1-52K, was decreased. While both short-read and long-read sequencing were performed on poly(A) selected RNA that is biased for detecting mature fully processed transcripts, low levels of incompletely processed RNA can be detected. Since nanopore sequencing starts at the 3′ cleavage and polyadenylation site, we were able to uniquely bin all reads that were cleaved at any of the L1-L5 late gene polyadenylation sites and assay for the presence of intron retention upstream of the first splice acceptor (Fig. S7a). Comparing the METTL3 KO to Wildtype cells or siMETTL3 to siCTRL data revealed evidence of increased intron retention/poor RNA processing in every late transcriptional unit (Fig. S7b). Overall, these data highlight that METTL3 promotes the expression and splicing efficiency of the alternatively spliced viral late transcriptional unit.

Utilizing the power of our long-read dRNA sequencing approach to map m6A sites to unique RNA isoforms, we asked if any specific features of these RNAs correlated with the loss of expression we observed after METTL3 knockdown. The total number of potential m6A sites per transcript was not correlated with change in expression (Spearman $\rho = -0.38$, Fig. 7d), however the total number of splice sites per transcript and RNA expression were negatively correlated (Spearman $\rho = -0.75$, Fig. 7e). The strongest negative correlation with METTL3-dependent changes in expression was found to be the length of

the longest intron within a specific transcript (Spearman $\rho = -0.84$, Fig. 7f). Using high-depth short-read sequencing and existing gene annotations to ask similar questions of host cell transcripts upon METTL3 depletion, we found 958 genes increased and 1298 decreased greater than two-fold with an FDR of 0.05% upon siMETTL3 treatment. Focusing on these significantly changed genes, we saw no correlation with the number of m6A regions (defined by meRIP-seq), total number of exons, or the longest intron within a single transcript (data not shown). While these correlations might hold true for specific cellular genes that have features similar to the viral transcriptome (e.g., high expression levels and many alternative splice and polyadenylation sites), global correlations appear confounded by the many diverse roles m6A modification can have on any single transcript.

## Discussion

In this work we have identified sites of m6A methylation in both cellular and viral transcripts during infection with adenovirus. Using a conventional antibody-based approach to detect m6A, we discovered that both viral early and late kinetic classes of mRNA were modified. To identify m6A within overlapping viral transcripts, we devised a technique to predict METTL3-dependent methylation sites in full-length RNAs using direct RNA sequencing. This technique identified specific m6A sites at nucleotide resolution within m6A-enriched regions, and revealed adenovirus RNA isoform-dependent sites of methylation. Ultimately, we found that m6A addition in the alternatively spliced viral late transcripts was important for mediating the efficiency of splicing and accumulation of these messages leading to productive infection. The benefit of m6A-mediated splicing efficiency was directly correlated to the intron length of the transcript, implicating a role for m6A in processing of long transcripts with many alternative splicing choices. Taken together, this work demonstrates m6A-mediated control over the splicing of viral transcripts.

We also defined a new method for isoform-specific m6A detection within complex transcriptomes. Compared to conventional antibody-based approaches, direct detection of m6A modification in RNA provides many advantages. Focusing on adenovirus RNAs, we show that our technique is highly reproducible between biological replicates with a very low false-positive rate (here defined as candidates that did not map within five

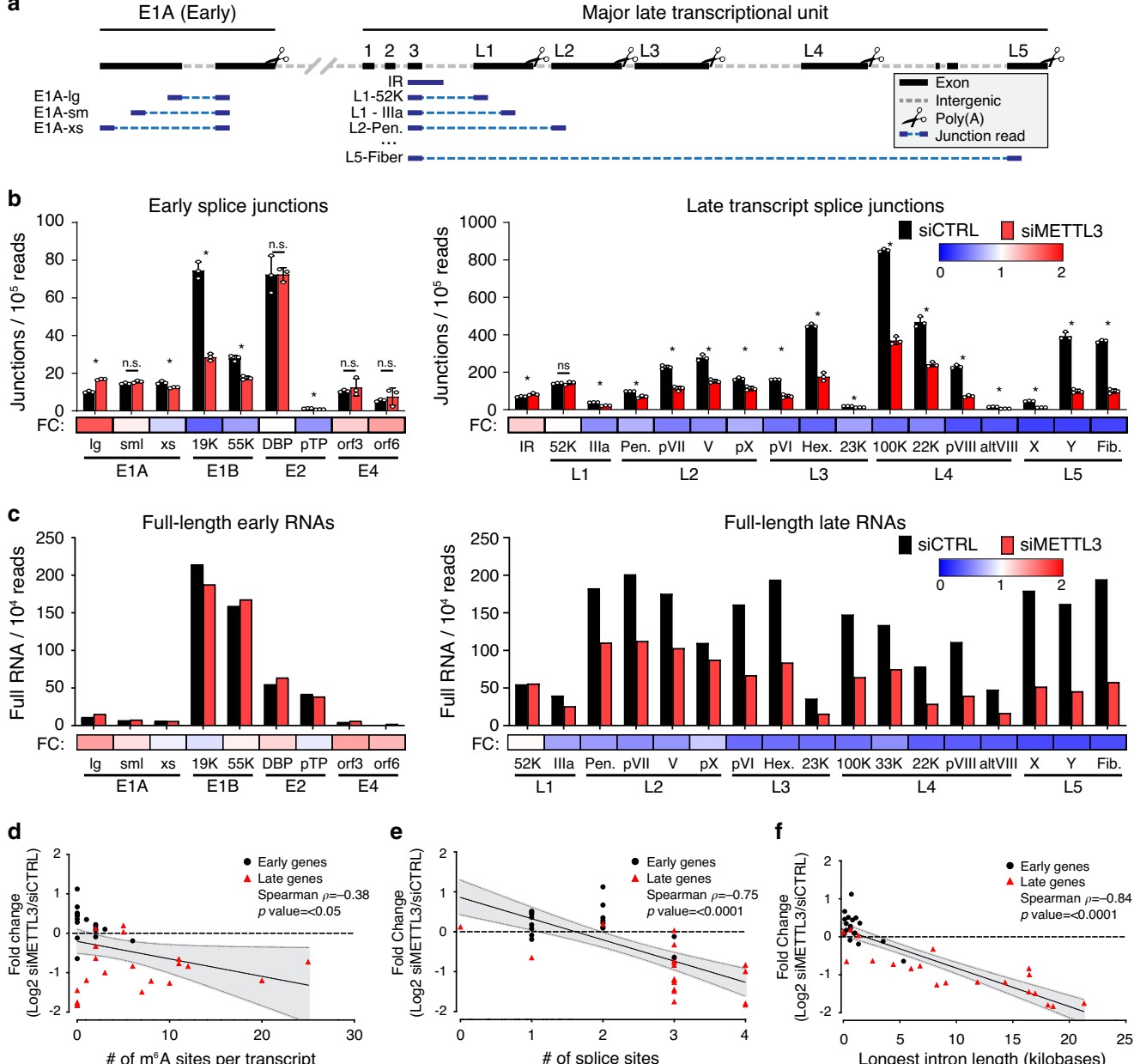

**Fig. 7 METTL3 knockdown globally dysregulates adenoviral late RNA processing. a** Schematic showing how junction-containing splice reads generated by Illumina sequencing can be used to predict specific transcript abundances when genes overlap. Short reads (dark blue) aligning specifically to viral exons (black) were filtered for the presence of a splice junction (dashed light blue line) that was only present in one viral transcript. **b** Splice junction containing reads indicative of adenovirus transcripts present in Illumina RNA-Seq data generated after infection of A549 cells where METTL3 was depleted by siRNA. Splice junction read depth was normalized to the total amount of reads mapping to both human and viral RNA per library. Early viral transcripts are shown on the left, MLP-derived late transcripts shown on the right. Fold-change (FC) between control siRNA and siMETTL3 for each transcript was plotted as a heatmap below the bar chart. Data depict three biological replicates with error bars showing standard deviation. For all assays significance was determined by unpaired two-tailed Student's *T*-test, where *$p \leq 0.05$ and ns = not significant. Graphs represent means +/− standard deviation. **c** Independently derived RNA was sequenced by ONT after METTL3 knockdown and adenovirus infection to yield full-length RNA sequences indicative of the labeled early and late viral transcripts. Fold-change (FC) between control siRNA and siMETTL3 for each transcript was plotted as a heatmap below the bar chart. **d** Individual viral transcripts are plotted as a function of log2 fold change of the siMETTL3/siCTRL data from Fig. 7c and the number of isoform-level m6A sites detected in Fig. 2a. Canonical early genes are coded with black squares and canonical late genes coded with red triangles, however the entire dataset was analyzed by Spearman's correlation test to yield a correlation rho (ρ) value and significance *p*-value. **e** Analysis performed as in **d**, but with log2 fold change and the total number of splice sites contained within each viral transcript. **f** Analysis performed as in **d**, but with log2 fold change and the length (in nucleotides) of the longest intron contained within each viral transcript. For panels **d**–**f** the black line represents the calculated best-fit linear regression, and the shaded gray area represents 95% confidence interval. Exact *p*-values are included in the source data file.

nucleotides of an AC dinucleotide). In fact, after candidate site masking, greater than 80% of all sites mapped directly to the adenine within an AC dinucleotide, with the rest often mapping within four nucleotides. This is consistent with the fact that nanopores read RNA in a five nucleotide window[61]. While we demonstrate this method by assaying for METTL3-dependent m6A modifications, theoretically this technique could work for other RNA modifications where the writing enzyme is known, such as methyl-5-cytosine and pseudouridine[44,45]. Compared to other single-nucleotide resolution approaches to m6A detection[48–50], direct RNA sequencing avoids the known biases associated with antibody targeting[51] and cDNA synthesis[42,49] and, unlike endoribonuclease cutting[50], is not specific to a small subset of methylated motifs. Currently, our technique works best with samples of high read depth, and is thus ideally suited to the study of viruses that dominate the RNA processing of their host cell. However, with improvements in nanopore sequencing technologies such as the PromethION which can yield much higher read depth, this technique should be scalable to the study of cellular transcripts as well.

The lack of long-read assays to map m6A methylation has hampered the identification of transcript isoform-specific modifications. While our method still does not enable single molecule resolution, the ability to aggregate similar isoforms in bulk using the long reads produced through direct RNA sequencing allows for isoform-level prediction of m6A sites. Indeed, we demonstrate that isoform-level identification was more sensitive for the detection of potential m6A sites in adenovirus mRNA than the aggregate exome-level data by as much as four-fold. While isoform-level alignment and filtering rely on quality annotations, this technique may need additional customization based on transcript structures in the target of choice. Furthermore, we have only shown the efficacy of this design on polyadenylated messenger RNA. Further studies focusing on non-adenylated or nascent RNAs should prove highly informative for defining the role of m6A-mediated regulation within introns and structural RNAs.

While the influence of m6A on splicing has been widely reported in both viral and cellular contexts[7,8,37,69], this finding has not been without controversy[70]. One study in particular showed that while m6A can be detected in nascent pre-mRNA, these locations do not change upon RNA maturation and nuclear export[15]. Furthermore, knockout of METTL3 in mouse embryonic stem cells led to no detectable change in cassette exon splicing[15]. In general, our findings agree with this as we do not see the global loss of any particular viral splicing event in the absence of METTL3, only a decrease in total amount of spliced transcripts. However, we did find that m6A modifications present in adenoviral late transcripts increase the splicing efficiency of these RNAs. This result is consistent with the results of others that have found that the presence of m6A near splice sites correlates with the rate and efficiency of splicing when studied in a time-resolved fashion in nascent RNA[71]. Furthermore, it is known that unspliced cellular RNAs resident in the nucleus can be targeted for decay by the nuclear exosome[68,72]. Therefore, our hypothesis is that efficient splicing requires m6A and leads to the accumulation of mature mRNAs that would otherwise be destroyed in the nucleus before accumulating in the cytoplasm.

The m6A modification has also been implicated in the regulation of alternative polyadenylation[47,62,69]. While the majority of the adenovirus early transcripts that were insensitive to METTL3 knockdown contain only one cleavage and polyadenylation site, the adenovirus late transcriptional unit contains five poly(A) sites, and usage must be regulated to be able to splice the downstream isoforms. It is intriguing to note that transcripts generated from the L1 region upstream of the first polyadenylation site were the viral late transcripts least affected by METTL3 knockdown. While our quantitative PCR reaction to test the splicing efficiency of the L5 region cannot fully rule out the possibility of fewer transcripts reaching this region due to altered poly(A) cleavage, our splicing reporter transgene assay that does not contain upstream poly(A) sites argues against this hypothesis. Splicing and polyadenylation are thought to happen co-transcriptionally[73], and any effect that m6A might have on coordinating these processes remains an interesting topic for future study.

In summary, we discovered that while both early and late adenoviral genes are marked by m6A, we only detected a loss of expression within late genes after depletion of METTL3. This was because m6A increased the efficiency of splicing within adenovirus late transcriptional units, where a plethora of potential splice sites leads to an abundance of alternative splice isoforms. Our data show that even when rates of transcription and RNA decay remain constant, increased splicing efficiency can lead to increased transcript abundance, presumably via protection against nuclear decay mechanisms that target unspliced transcripts. This work highlights how RNA modifications can regulate distinct stages of viral gene expression.

## Methods

**Cell culture**. All cell lines were obtained from American Type Culture Collection (ATCC) and cultured at 37 °C and 5% $CO_2$. All cell lines tested negative for mycoplasma infection and were routinely tested afterwards using the LookOut Mycoplasma PCR Detection Kit (Sigma-Aldrich). A549 cells (ATCC CCL-185) were maintained in Ham's F-12K medium (Gibco, 21127-022) supplemented with 10% v/v FBS (VWR, 89510-186) and 1% v/v Pen/Strep (100 U/ml of penicillin, 100 μg/ml of streptomycin, Gibco, 15140-122). HeLa cells (ATCC CCL-2) and HEK293 cells (ATCC CRL-1573) were grown in DMEM (Corning, 10-013-CV) were grown in DMEM supplemented with 10% v/v FBS and 1% v/v Pen/Strep.

**Viral infection**. Adenovirus serotype 5 (Ad5) was originally purchased from ATCC. All viruses were expanded on HEK293 cells, purified using two sequential rounds of ultracentrifugation in CsCl gradients, and stored in 40% v/v glycerol at −20 °C (short term) or −80 °C (long term). Viral stock titer was determined on HEK293 cells by plaque assay, and all subsequent infections were performed at a multiplicity of infection (MOI) of 20 PFU/cell. Cells were infected at 80-90% confluent monolayers by incubation with diluted virus in a minimal volume of low serum (2%) F-12K for 2 h. After infection viral inoculum was removed by vacuum and full serum growth media was replaced for the duration of the experiment.

**Plasmids, siRNA, and transfections**. Fiber-Transgene constructs were created from the previously generated m6A-null psiCheck2 reporter plasmid[34], where both Firefly and Renilla luciferase genes have all m6A-DRACH motifs ablated by silent mutations. The original 3'UTR reporter was removed from the end of Renilla luciferase using XhoI and NotI restriction enzymes and this site was healed by the insertion of a recombinant multiple cloning site containing XhoI, AgeI, and NotI using two annealed DNA oligos with complementary sticky ends (Obtained from IDT). From this reporter construct lacking significant 3'UTR, the chimeric intron downstream of the SV40 promoter and upstream of Renilla luciferase was excised using StuI and NheI. Using compatible cut sites, a DNA fragment was cloned into this site encoding the third exon of the Ad5 late gene unit (nucleotides 9,644 to 9,733), all intervening intron that did not contain other Ad5 genes or splice sites (nucleotides 9733 to 10,631 and 30,867 to 31,042), and the first 482 nucleotides of the Fiber exon and ORF (31,042 to 31,524) such that the Fiber ORF would continue in frame to the Renilla luciferase ORF (WT Fiber). Alternatively, the same fragment but with all 15 DRACH motifs silently mutated was used to generate m6A Mut Fiber. Both fragments were excised from plasmids created as a PriorityGENE service from GeneWiz from provided sequences. All DNA oligos can be found in Supplementary Table 1. DNA transfections were performed using the standard protocol for Lipofectamine2000 (Invitrogen).

The following siRNA pools were obtained from Dharmacon: non-targeting control (D-001206-13-05), METTL3 (M-005170-01-0005), METTL14 (M-014169-00-0005), WTAP (M-017323-02-0005), YTHDC1 (M-015332-01-0005), FTO (M-004159-01-0005), and ALKBH5 (M-004281-01-0005). The following siRNAs were obtained from Qiagen: YTHDF1 (SI00764715), YTHDF2 (SI04174534), YTHDF3 (SI04205761), FTO (SI04177530), and ALKBH5 (SI04138869). All siRNA transfections were performed using the standard protocol for Lipofectamine RNAiMAX (Invitrogen). Poly(I:C) was provided pre-complexed with transfection

reagent (Invivogen tlrl-piclv), and was reconstituted fresh with molecular grade water before added to cells at a concentration of 500 ng/ml.

**METTL3 knockout**. Since METTL3 has previously been reported to be an essential gene, a rescue cell line was pre-constructed that contained a CRISPR-insensitive METTL3 transgene under a tetracycline-inducible promoter. In brief, the 2X-Flag Tagged METTL3 transgene[32] was cloned from pEFTak into the BB726[74] entry vector and the NGG Cas9 PAM site at nucleotide position 117 was silently mutated (G to C) via the Stratagene QuickChange Site-Directed Mutagenesis protocol (primer sequence in Supplementary Table 1). This plasmid was then co-transfected into A549 HiLo cells with a transiently expressed Cre Recombinase before being selected by puromycin[74]. This allows integration of METTL3 into a conserved tetracycline-regulated genomic locus, and expression was tested both by immunoblot and immunofluorescence for selection efficiency.

GFP-Cas9 expressing plasmid pX330[75] was constructed to contain the single guide RNA (Supplementary Table 1) from the GECKO CRISPR library. The A549-METTL3-HiLo cell line was then induced to express Cas9-insensitive METTL3 for one day before transfected with the single pX330 plasmid containing both GFP-Cas9 and METTL3 sgRNA. Twenty-four hours post transfection, GFP expressing cells were sorted as single cells by fluorescence-activated cell sorting (FACS) into 96-well plates for clonal expansion. Tetracycline was refreshed biweekly to maintain transgene expression for the entire outgrowth until resulting cell lines reached large enough numbers to be viably frozen. Afterwards, tetracycline was withdrawn and cells were cultured for a further two weeks before the presence of endogenous METTL3 was assayed for by both immunoblot and immunofluorescence. Two clones were picked that displayed undetectable levels of endogenous METTL3 at the single cell level, and Cas9-mediated lesions were detected by Sanger sequencing of a PCR amplicon derived from genomic DNA (PureLink Genomic DNA kit, Invitrogen).

**meRIP-Seq and meRIP-qPCR**. A549 cells in 10 cm plates were lysed directly in TRIzol (Thermo Fisher) and RNA was extracted and DNaseI treated (Qiagen). Poly (A) RNA was selected with the Poly(A)Purist MAG Kit before being fractionated (Thermo Fisher) with RNA Fragmentation Reagents (Thermo Fisher) and ethanol precipitated. For meRIP-qPCR the following protocol was identical but RNA was not poly(A) purified or fragmented. MeRIP was performed using the EpiMark N6-methyladenosine Enrichment Kit (NEB) following manufacturer's instructions. The following modifications were made following the previously published protocol[32]. Immunoprecipitations were washed with low and high salt wash buffers and RNA was eluted from the beads with 5 mM m$^6$A salt (Santa Cruz Biotechnology).

For meRIP-Seq RNA-seq libraries were prepared from both eluate and 10% input mRNA using the TruSeq mRNA library prep kit (Illumina), subjected to quality control (MultiQC), and sequenced on the HiSeq4000 instrument. Both the IP and input samples were mapped to the GRCh37/hg19 genome assembly and the Ad5 genome using the RNA-seq aligner GSNAP[76] (version 2019-09-12) or STAR (v2.5.4b)[77]. We called peaks using MACS2 (v2.1.2)[78] with the following flags: "-q 0.05 -B –call-summits –keep-dup auto –nomodel –extsize 150". HOMER (v4.11)[79] was then used to identify motifs enriched in the identified m$^6$A-peaks. Metagene analysis and visualization was done using deepTools2 (v3.3.1)[80].

**Antibodies, immunoblotting, and immunofluorescence**. The following primary antibodies were used for cellular proteins: METTL3 (Novus Biologicals H00056339, WB: 1:400, IF: 1:50), METTL14 (Sigma-Aldrich HPA038002, WB: 1:5000, IF: 1:100), WTAP (Proteintech 60188, WB: 1:400, IF: 1:100), YTHDC1 (Abcam ab122340, WB: 1:2000, IF: 1:100), YTHDF1 (Abcam ab99080, WB: 1:500, IF: 1:100), YTHDF2 (Proteintech 24744-1-AP, WB: 1;1000, IF: 1:100), YTHDF3 (Sigma-Aldrich SAB2102735, WB: 1:500), FTO (Abcam ab92821, WB: 1:300, IF: 1:100), ALKBH5 (Sigma-Aldrich SAB1407587, WB: 1:250, IF: 1:100), β-Actin (Sigma-Aldrich A5441-100UL, WB 1:5000), GAPDH (GeneTex 41577, WB:1:20,000), and RNA Pol II p-Ser2 (Abcam ab5095, IF: 1:400). Primary anti-Flag tag antibody was obtained from Sigma-Aldrich (F7425-.2MG, WB 1:2000). Primary antibodies against viral proteins were obtained from: rabbit polyclonal against adenovirus Hexon, Penton, and Fiber (Gift from J. Wilson, WB 1:10,000), mouse anti-DBP (Gift from A. Levine, Clone: B6-8, WB 1:1000, IF 1:400), polyclonal rabbit anti-DBP (Gift from A. Levine, IF 1:40,000), mouse anti-E1A (BD 554155, WB: 1:500), and mouse anti-E1B55K (Gift from A. Levine, Clone: 58K2A6, WB 1:500).

For western immunoblotting protein samples were prepared by directly lysing cells in lithium dodecyl sulfate (LDS) loading buffer (NuPage) supplemented with 1% beta-mercaptoethanol (BME) and boiled at 95 °C for 10 min. Equal volumes of protein lysate were separated SDS-PAGE in MOPS buffer (Invitrogen) before being transferred onto a nitrocellulose membrane (Millipore) at 35 V for 90 min in 20% methanol solution. Membranes were stained with Ponceau to confirm equal loading and blocked in 5% w/v non-fat milk in TBST supplemented with 0.05% w/v sodium azide. Membranes were incubated with primary antibodies in milk overnight, washed for three times in TBST, incubated with HRP-conjugated secondary (Jackson Laboratories) for 1 h and washed an additional three times in TBST. Proteins were visualized with Pierce ECL or Femto Western Blotting

Substrate (Thermo Scientific) and detected using a Syngene G-Box. Images were processed and assembled in Adobe Photoshop and Illustrator CS6.

For immunofluorescence A549 cells were grown on glass coverslips in 24-well plates, mock-infected or infected with Ad5 for 18 h, washed twice with PBS and then fixed in 4% w/v Paraformaldehyde for 15 min. Cells were permeabilized with 0.5% v/v Triton-X in PBS for 10 min, and blocked in 3% w/v BSA in PBS (+0.05% w/v sodium azide) for 1 h. Primary antibody dilutions were added to coverslips in 3% w/v BSA in PBS (+0.05% w/v sodium azide) for 1 h, washed with 3% BSA in PBS three times, followed by secondary antibodies and 4,6-diamidino-2-phenylindole (DAPI) for 1 h. Secondary antibodies were used at 1:500 dilution and conjugated to Alexa-Fluor 488 (Invitrogen A-11001 or A-11008), 555 (Invitrogen A-21422 or A-21428), or 568 (Invitrogen A-11004 or A-11011) in anti-mouse or anti-rabbit. Coverslips were mounted onto glass slides using ProLong Gold Antifade Reagent (Cell Signaling Technologies). Immunofluorescence was visualized using a Zeiss LSM 710 Confocal microscope (Cell and Developmental Microscopy Core at UPenn) and ZEN 2011 software. Images were processed in FIJI (v1.52p) and assembled in Adobe Photoshop and Illustrator 2020.

**Inhibitors and small molecules**. 3-Deazaadenosine was purchased from Sigma-Aldrich (D8296) and resuspended in Dimethyl Sulfoxide (DMSO) before being added to cells at the time of infection to a final concentration of 25 μM. Ruxolitinub was purchased from Sigma-Aldrich (G-6185) and resuspended in DMSO before being added to cells immediately prior to infection with Ad5 or transfection with poly(I:C) at a final concentration of 4 μM. 5,6-dichloro-1-β-D-ribofuranosyl-1H-benzimidazole (DRB) was purchased from Cayman Chemical (Item No 10010302) and resuspended in DMSO before being added to cells at 24 h post infection to a final concentration of 60 μM.

**RNA isolation and qPCR**. Total RNA was isolated from cells by either TRIzol extraction (Thermo Fisher) or RNeasy Micro kit (Qiagen), following manufacturer protocols. RNA was treated with RNase-free DNase I (Qiagen), either on-column or after ethanol precipitation. RNA was converted to complementary DNA (cDNA) using 1 μg of input RNA in the High Capcity RNA-to-cDNA kit (Thermo Fisher). Quantitative PCR was performed using the standard protocol for SYBR Green reagents (Thermo Fisher) in a QuantStudio 7 Flex Real-Time PCR System (Applied Biosystems). All primers were used at 10 μM and sequences can be found in Supplementary Table 1. All values were normalized by the ΔΔCt method by normalizing first to internal controls such as *HPRT1* and *GAPDH*.

**Viral DNA replication qPCR**. Infected cells were harvested at the indicated time points post infection by trypsin and total DNA was harvested using the PureLink Genomic DNA kit (Invitrogen). DNA quantity was assessed by qPCR and SYBR green reagents using primers for genomic regions of Ad5 and normalized to cellular tubulin (See Supplementary Table 1 for primer sequence). Entry of the viral genome was assessed at the 4 h infection time point, and all subsequent values were normalized to this by the ΔΔCt method.

**Plaque assays**. Infected cells seeded in 12-well plates were harvested by scraping at the indicated time points and lysed by three cycles of freeze-thawing in liquid nitrogen. Cell debris was removed from lysates by centrifugation at max speed (21,130 × *g*), 4 °C, 5 min. Lysates were serially diluted in DMEM supplemented with 2% v/v FBS and 1% v/v Pen/Strep to infect a confluent monolayer of HEK293 cells seeded in 12-well plates. After incubation for 2 h at 37 °C, the infection media was removed, and cells were overlaid with DMEM containing 0.45% w/v SeaPlaque agarose (Lonza) in addition to 2% v/v FBS and 1% v/v Pen/Strep. Plaques were stained using 1% w/v crystal violet in 50% v/v ethanol between 6 to 7 days post-infection.

**Metabolic labeling of RNA for transcription rate determination**. To assess relative RNA transcription rate, cells were treated with 1 mM 4-thiouridine (4sU; Sigma T4509) for exactly 10 min. Infection was stopped and RNA harvested using 1 ml TRIzol (Thermo Fisher), following manufacturer's instructions. A fraction of the total RNA was reserved as input, and the remaining 4sU-labeled nascent RNA was biotinylated using MTSEA-Biotin-XX (Biotium, 90066) as previously described[67,81]. Nascent RNA was separated from unlabeled RNA using MyOne C1 Streptavidin Dynabeads (Thermo Fisher Scientific, 65-001), biotin was removed from nascent RNA using 100 mM dithiothreitol (DTT), and RNA was isopropanol precipitated. One μg of total RNA (T) and an equivalent volume of nascent RNA (N) were converted to cDNA and qPCR was performed as described above. Relative transcription rates were determined by the ΔΔCt method to compare nascent transcript levels between control and siRNA treated cells normalized to nascent *GAPDH* RNA.

**RNA-sequencing**. Total RNA from three biological replicates of Control knockdown or three biological replicates of METTL3-knockdown A549 cells infected with Ad5 for 24 h were sent to Genewiz for preparation into strand-specific RNA-Seq libraries. Libraries were then run spread over three lanes of an Illumina HiSeq 2500 using a 150 bp paired-end protocol. Raw reads were mapped to the GRCh37/

hg19 genome assembly and the Ad5 genome using the RNA-seq aligner GSNAP[76] (version 2019-09-12). The algorithm was given known human gene models provided by GENCODE (release_27_hg19) to achieve higher mapping accuracy. We used R package ggplot2 (v3.3.0) for visualization. Downstream analysis and visualization was done using deepTools2 (v3.3.1)[80]. Splice junctions were extracted using RegTools (v0.2.0)[82] and visualized in Integrative Genomics Viewer.

**Nanopore direct RNA sequencing.** Direct RNA sequencing libraries were generated from 680 to 1800 ng of poly(A) RNA, isolated using the Dynabeads™ mRNA Purification Kit (Invitrogen, 61006). Isolated poly(A) RNA was subsequently spiked with 0.3 μl of a synthetic Enolase 2 (ENO2) calibration RNA (Oxford Nanopore Technologies Ltd.) and prepared for sequencing as described previously[56,61] Sequencing was performed on a MinION MkIb using R9.4.1 (rev D) flow cells (Oxford Nanopore Technologies Ltd.) for 18–23 h (one library per flowcell) and yielded between 1,000,000 – 2,000,000 reads per dataset. Raw fast5 datasets were then basecalled using Guppy v3.2.2 (-f FLO-MIN106 -k SQK-RNA002) with only reads in the pass folder used for subsequent analyses. Sequence reads were aligned against the Ad5 genome, using MiniMap2 (v2.15 -ax splice -k14 -uf –secondary = no), a splice aware aligner[64], with subsequent parsing through SAMtools (v1.9)[83] and BEDtools (v2.27.1)[84]. Here sequence reads were filtered to retain only primary alignments (SAM FLAG 0 (top strand) or 16 (bottom strand)). Coverage plots (Figs. 2 and 4) were generated using the R packages Gviz (v3.10)[85] and GenomicRanges[86]. Isoform-level analysis was performed by aligning sequence reads with MiniMap2 to an Ad5 transcriptome that we recently re-annotated[63]. Subsequent parsing with SAMtools was used to filter out sequence reads that could not be unambiguously assigned to a single transcript. Here, only primary alignments (SAM flag 0) with mapping qualities (MapQ) >0 were retained. Unambiguous transcripts were defined as those that began at a designated 3′ poly(A) tail and extended to within 50 nucleotides of an annotated TSS, or within the third exon of the late transcripts tripartite leader.

**Nucleotide resolution analysis of m[6]A sites using direct RNA sequencing.** The identification of putative m[6]A sites on adenoviral RNAs was performed using a new tool, DRUMMER (https://github.com/DepledgeLab/DRUMMER/), which predicts RNA modifications based on comparative profiling of basecall error rates at both exome and isoform level. Briefly, DRUMMER takes sorted.bam files from two datasets as input and parses these individually in a stranded manner using BAM Readcount (https://github.com/genome/bam-readcount) to generate base-call distributions (i.e., the number of A, C, G, U, and indels) at each position in the alignment. For each comparison (e.g., M3KO versus WT), a G-test was performed on the base-call distributions at each position in each dataset with subsequent Bonferroni correction for multiple testing. Positions with a read coverage of <100× in one or both datasets being compared were excluded from this analysis. To further filter the dataset following the G-test, we calculated the ratio of match: mismatch fractions at each position. Candidate sites were those that gave a statistically significant G-test result (bonferroni adjusted $p < 0.01$) and had a one-fold or higher reduction in the match:mismatch fraction in the WT dataset compared to the METTL3 knockout dataset. Subsequent masking of clustered candidates was performed by collapsing all sites located within five nucleotides of the same AC motif into a single representative, chosen by picking the site with highest G-test score. Where sites were located within ten nucleotides of each other, but further than five nucleotides from an AC motif, collapsing was performed by selecting the site with the highest G-test score.

To further validate our data, we also performed m[6]A site prediction using NanoCompore, which predicts RNA modifications based on signal level analyses of current intensity and dwell time for nucleotides translocating through nanopores[65]. Here, read indexing and resquiggling were performed using Nanopolish and NanopolishComp prior to running NanoCompore SampComp (–logit –sequence_context 2 –sequence_context_weights harmonic) to compare both biological replicates of the WT and METTL3-knockout datasets in a 2 × 2 analysis. Note that NanoCompore processing times were extremely slow for transcripts with >1000 dRNA-Seq reads assigned, so a random subsampling approach (1000 reads) was used in these situations. Putative m[6]A modified sites were identified on the basis of a GMM logit p-value (context 2) <0.01 and an Logit LOR score >0.5 or < −0.5. Where multiple sites were present with five nucleotides, only the site with the lowest p-value was retained. To calculate the degree of methylation in WT and METTL3-knockout datasets, we parsed the cluster counts column to determine the fraction of reads in datasets with an m[6]A at a defined sequence position with the m[6]A depletion estimate representing the difference between these values.

**Reporting summary.** Further information on research design is available in the Nature Research Reporting Summary linked to this article.

## Data availability

Fast5 (Nanopore) and fastq (Illumina) datasets generated as part of this study can be downloaded from the European Nucleotide Archive (ENA) under the following study accession: PRJEB35652. The authors declare that all other data supporting the findings of this study are available within the article and its Supplementary Information files, or are available from the authors upon request. Source data are provided with this paper.

## Code availability

All code pertaining to detection of m[6]A sites via direct RNA Sequencing (DRUMMER) is available at https://github.com/DepledgeLab/DRUMMER/.

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

## Acknowledgements

We extend special thanks to Ian Mohr (New York University School of Medicine) for his many insightful comments and for support of DPD in part through National Institutes of Health (NIH) grants R01-AI073898 and R01-GM056927. We also thank members of the Weitzman and Mohr/Wilson Labs for insightful discussions and input. We are grateful to A. Berk, P. Branton, G. Ketner, A. Levine, and D. Ornelles for generous gifts of reagents. We thank our colleagues Kristen Lynch, Fange Liu, and Peter Choi for helpful suggestions and careful reading of the manuscript. We also thank Tommaso Leonardi (Istituto Italiano di Tecnologia) for insightful discussions regarding NanoCompore. We thank the UPenn Cell and Developmental Biology Microscopy Core for imaging assistance. This work was supported through NIH grants R21-AI130618 and R21-AI147163 (ACW), R01-AI125416 and R21-AI129851 (S.M.H. and C.E.M.), and R01-AI145266, R01-AI121321, and R01-CA097093

(MDW). S.M.H. was supported by the Burroughs Wellcome Fund (17PRE336700170). N.S.G. was supported by an American Heart Association Predoctoral Fellowship. ABRM was supported by the Natural Sciences and Engineering Research Council of Canada. J.S.A. was supported by the Sackler Institute of Graduate Biomedical Sciences. Additional support came from the NCI T32 Training Grant in Tumor Virology T32-CA115299 (A.M.P.) and Individual National Research Service Award F32-AI138432 (AMP).

## Author contributions

A.M.P., D.P.D., and M.D.W. conceived of the project and designed the experiments; C.E.M., A.C.W., and S.M.H. provided additional input into study design; A.M.P., N.S.G., A.N.D.F., and D.P.D. performed the experiments; D.P.D. designed the nanopore m6A detection approach; J.S.A. codified the DRUMMER pipeline; K.E.H., A.B.R.M., and D.P.D. performed computational analyses; A.M.P. and D.P.D analyzed all additional data; A.M.P., D.P.D., and M.D.W. wrote the manuscript; All authors read, edited and approved the final paper.

## Competing interests

C.E.M. is a cofounder and board member for Biotia and Onegevity Health, as well as an advisor or compensated speaker for Abbvie, Acuamark Diagnostics, ArcBio, BioRad, DNA Genotek, Genialis, Genpro, Karius, Illumina, New England Biolabs, QIAGEN, Whole Biome, and Zymo Research. The remaining authors declare no competing interests.
