## [Peer Review File · Nature Communications]

Reviewers' comments:

Reviewer #1 (Remarks to the Author):

In this paper, Price et al. use cutting-edge techniques to catalog the m6A methylation events across the adenovirus transcriptome and they explore the functional importance of those methylation events. Given the emerging roles of m6A in gene expression in viruses and their hosts, these studies are timely and important for the field. Overall, this is a strong manuscript. The adenoviral m6A analysis in Figures 2-4 are particularly rigorous and clearly presented. Furthermore, the observation that adenovirus recruits the m6A machinery to viral replication centers is interesting. The authors further explore the effects of m6A on viral transcript accumulation and processing and propose that methylation of viral transcripts is required for late gene splicing. They present reasonable data supporting this conclusion, but their interpretations are currently too strong given the data presented.

In particular, I do not think that the authors have ruled out the contribution of m6A to RNA decay in their system. The only assay addressing RNA degradation is a calculated half-life based on nascent RNA and input levels. Given the importance of this conclusion, it is essential to complement this calculation with a more direct (e.g. transcription shut-off) assay.

Moreover, I am not sure whether the half-life calculation is applicable to adenovirus infection. I think these assays assume the total RNAs have reached steady-state. If so, that assumption is not valid during viral infection where late gene transcripts continue to accumulate over time throughout infection. For example, the Fiber RNA continues to increase over the course of infection from 24-30hpi (Crisostomo et al 2019). If this is driven by changing transcription levels, then the assumption of constant transcription does not hold (ref 80). On the other hand, if RNA accumulates simply due to decay rate being slower than a constant transcription rate, then the value assigned to input will be different if the samples were taken at 24 hpi compared to 30 hpi. The resulting half-life determinations would reflect this difference, but this distinction has nothing to do with changes in decay rates. I'm not an expert in decay calculations, so my reasoning may be faulty. Intuitively, however, it seems to present an issue in the analysis. Either way, it is important to use an orthogonal approach because the downstream interpretations completely depend on the conclusion that decay is unaffected.

1. The reporter assay (Fig 6h-i) is potentially powerful and should be expanded upon because this is the only data supporting a direct role for methylation as opposed to indirect consequences of cellular responses to METTL3 knockdown. Several assays should also be relatively straightforward to perform that would strengthen their conclusions:

a. For the reporter assay and the other spliced/unspliced assays (Fig 6d-f), they should show the levels of the unspliced and spliced RNAs in the supplemental. This is important because if changes in the ratios are driven solely by decreased spliced RNA, this result is equally consistent with a splicing defect or a decay defect.

b. The siMETTL3 in the mutant construct decreases, even though it's not statistically significant. Therefore, it is difficult to discern whether it's not significant because there's no change in levels or

because of experimental variability. Even the WT samples only decrease by ~25%, so the decrease shown may be closely mirroring this effect. Showing the individual values as dots on the graph would help interpretation of these data and more replicates may also clarify the interpretations.

c. Examination of intronless WT/Mut reporters would go a long way in solidifying the conclusion that splicing drives the changes in RNA abundance.

d. Conversely, a reporter with a strongly spliced m6A-independent intron would be interesting to test.

e. They should perform the YTHDC1 knockdowns to the reporter to further substantiate their other mechanistic data.

2. Fig 7. The fully processed full-length RNAs decrease in siMETTL3 samples, so the spliced junction reads will also go down regardless of the mechanism. If the decreased splice junctions go down and a concomitant increase in IR is observed, the case for splicing is much stronger. However, only a very small increase in IR is observed. One could reasonably argue that the unspliced IR transcripts are quickly degraded. Alternatively, one could argue that the mRNA is less stable in the siMETTL3 samples. Therefore, the result is ambiguous with respect to mechanism. In addition, browser shots comparing siCtrl to siMETTL3 would be nice to see to help readers visually inspect the reads on the introns/exons.

Reviewer #2 (Remarks to the Author):

In this work, Price et al. used direct RNA long-read sequencing to precisely map m6A sites on the complex transcriptome of Adenovirus. The authors show that transcripts of both early and late viral genes are modified, but report a differential effect of m6A, where reduction in m6A affects mostly late but not early viral gene expression. Mechanistically, the author show that m6A is important for proper splicing of late adenovirus genes. In general the experiments are well executed and the manuscript is well written.

Below are few suggestion that can help supporting the conclusions and improving the clarity of the manuscript

The most substantial improvement can be if the authors will use their long read and short read m6A sequencing data to strengthen the direct connection between the splicing defect and m6A modification. Is the effect of METTL3 depletion on late gene expression is correlated or restricted to modified transcripts? Are the genes that are changing contain m6A? can the authors show that other transcripts that are less affected (like L1-52K) are not (or less) modified? Is the defect in splicing related to length of the introns? Number of splice sites?

Specifically, in figure 7C the authors use the long reads to show there is a general reduction in late viral gene expression but are the long reads also support there are more non-spliced transcripts ? Since this data should allow to directly connect the RNA-isoform to m6A sites, can the authors show that un-spliced RNA contain less m6A signal than their spliced counterparts?

Another important point is whether the defect in Splicing is limited to the virus? If the effect also seen in

other cellular genes (with many splicing acceptor sites?) is there a general correlation between m6A and splicing defect? Or any other feature that correlate with splicing defect and can give some insights about about the molecular mechanism.

Minor points:

1. Fig. 6f: can the authors also show a decrease in the viral titer following knockdown of YTHDC1?
2. Fig. 1b+c: in this figure, the authors show the localization of m6A writer and nuclear reader, as well as RNA Pol II, around the viral DNA replication sites (marked by DNA binding protein (DBP)). The text suggest these m6A machinery components are recruited to the replication sites, similarly to other cellular factors during viral infection. However, this localization of m6A components might be a result of their (known) association with RNA pol II and not of an active recruitment to these factors by the virus.
3. Fig. 2d – why were GAPDH and HPRT chosen as controls? The meRIP-seq data shows that they are not modified?
4. Fig. 6c: the primers for E1A spliced isoform, as it appears in the illustration, might also amplify the un-spliced transcript (the intron is short enough). It would be better to use a fw primer which sits on the exon-exon junction or at least explain how this option was excluded
5. Fig. 6a: What are the values in this graph normalized to? The siControl of E1A is not 100%.
6. Fig. S4e: what is the difference between the two controls in infected cells?
7. Fig. 7b and c: please add significance asterisks to all bars, the current presentation is quite confusing.

Reviewer #3 (Remarks to the Author):

In this work, Price et al investigate the role of m6A RNA modifications in adenoviral transcripts. This is a timely and interesting topic, and the authors address it with a nice combination of tools and technologies. To address the role of m6A modifications during adenoviral infection, the authors performed transcriptome-wide mapping using MeRIP-seq and direct RNA sequencing (DRS), and in the case of DRS, combined with the use of METTL3 KO cells. The main findings of the authors are: 1) that METTL3 is the enzyme responsible for placing m6A RNA modifications in adenoviral RNAs; 2) adenoviral m6A can be identified using DRS at transcript level and single nucleotide resolution and 3) that m6A in adenoviral RNAs is important for splicing of late adenoviral transcripts, but not for early ones. However, based on the data presented in the manuscript unfortunately would need additional analyses/experiments/clarifications to support the claims. With regards to claim #1, the authors indirectly show that METTL3 knockdown causes decrease in expression of 2 viral transcripts using MeRIP-qPCR (Figure 2d); however, this does not necessarily mean that the viral m6A is placed by METTL3. Claiming that METTL3 is the enzyme responsible for adenoviral methylation is a major finding of the manuscript, however they only show this indirectly using MeRIP-qPCR, and considering that this is one of the major findings of the work, it should be validated using an additional technique such as mass spectrometry or immunonorthern of viral transcripts when infecting METTL3 KO cells vs control cells. With regards to claim #2, the first method (MeRIP-Seq) led to broad peaks, and the data was not analyzed in the context of viral RNA. Therefore, all results rely on the DRS analysis. However, the authors

should try to compare DRS to other well-established techniques such as MeRIP-Seq (i.e. to repeat their first experiment, but also sequencing the METTL3 KO strain, which wasn't done), or use miCLIP (which produces single nucleotide resolution) instead of MeRIP-Seq. Moreover, it would be recommendable to compare their DRS predicted sites to those predicted using other DRS m6A prediction software, such as Tombo, EpiNano or NanoCompore, and see if their predicted sites (and consequently their conclusions) hold. With regards to claim #3, the authors suggest that m6A only affects splicing of late transcripts, but it is unclear whether this effect might be only seen in late transcripts because the authors used for this experiment siRNA-METTL3, rather than the METTL3 KO cells. Could it be that the siRNA takes time to show its effect and thus, only late viral transcripts are affected? Overall, this work is an important contribution to the field, and tackles very relevant and interesting questions; however, additional experiments/analyses/clarifications are needed to support the main claims of the paper.

Major Comments

- 1) Which is the expected abundance of m6A based on previous works? Which is the observed abundance based on the current work? Are the results obtained direct RNA sequencing data in agreement with previous m6A abundance results, in terms of abundance of the m6A modification?
- 2) The authors mention that meRIP-Seq was not ideal because it produced broad peaks. Why was this? Did the authors try to predict motifs within their peaks? Could the authors try a different peak-calling algorithm? What was the size of the peak in cellular RNA transcripts compared to viral RNA transcripts? Other works have shown that m6A RIP-Seq could be done on viral RNA, as the authors mention in their Introduction. Could the authors perhaps try miCLIP (instead of MeRIP-Seq) on METTL3 KO cells to avoid the broad peaks? It would be key to validate the predicted m6A sites (main claim of the paper) by showing a consistency between DRS predicted m6A and well-established orthogonal methods to DRS based on Illumina, such as MeRIP-Seq, miCLIP or MaZTER-Seq. On the other hand, the authors performed MeRIP-Seq in infected and uninfected cells, but not in METTL3-KO cells. Wouldn't that help identify which of the "broad peaks" are in fact non-specific, and identify METTL3-specific peaks? This is indeed the design the authors did for DRS, which succeeded.
- 3) A key claim from this work is that METTL3 is the methylase responsible for methylating adenoviral RNA. The authors show that METTL3 knockdown causes decrease in expression of viral transcripts (Figure 2d); however, to my understanding, this does not necessarily mean that the viral m6A is placed by METTL3. An example of this is shown by the authors in Fig.5A, where the authors show that siMETTL3 doesn't lead to decreased DBP, despite DBP contains m6A sites predicted by their DRS. Thus, the meRIP-qPCR data shown is insufficient to draw the conclusion that METTL3 is responsible for viral RNA modifications. Additional validation is needed in this specific point because it is one of the major points of the manuscript. One possibility would be to measure m6A levels of the viral RNAs using mass spectrometry or immunonortherns of viral-infected parent vs viral-infected METTL3 knockout strains.
- 4) Regarding the DRS data, the authors collapsed NNACN sites and focus on the analysis of these sites

for the rest of the paper - but what about the remaining sites that don't meet the motif? Were there other sites predicted as "different" using the G-test after bonf correction that did not meet this motif? What proportion of 5-mers that passed the test in fact met the NNACN motif? Could it be that perhaps motifs that do not fall into this motif are actually the most abundant? The authors show in Figure 3C the distance of sites to NNACN data, but that is already after all filtering and masking. I think would be good to determine whether other sites that are not NNACN might indeed be highly represented among the k-mers that are depleted in METTL3-KO vs WT cells, i.e. to do the analysis "blind to the motif", rather than "guided by the motif". The motif should appear as a consequence, if it is the true motif. Could it be that the DBP m6A sites for example (whose expression was unaffected by METTL3 KO), are actually METTL3-independent?

5) Regarding the nanopore DRS data analysis, the authors use a specific pipeline which employs mismatch errors followed by a G-test. While this is theoretically correct, it is unclear how this method would compare to others. It would be good if the authors would show how would their predicted m6A sites look like if they had used other methods such as Nanocompore, EpiNano or Tombo. Which of them are robustly predicted by at least 2 out of 4 algorithms, for example? Would the conclusions of the work hold if the "m6A sites" would be those that are predicted in common by at least 2 different algorithms?

6) The authors show in Figure 7 that METTL3 knockdown globally dysregulates adenoviral late RNA processing but not early one. This result is very interesting, but to my understanding, was only conducted at one specific time point (t=24hpi) and using siRNA-METTL3, not METTL3 KO cells. Could these results be reflecting the fact that siMETTL3 takes a time until it has an effect, only causing the effect in splicing in late viral transcripts, but not in early viral transcripts? How can we outrule the possibility that siRNA did not affect early transcripts because there was still sufficient METTL3 mRNAs and proteins when the early viral particles were produced? If this possibility cannot be outruled, could then the authors perhaps confirm this by doing the experiment using METTL3 KOs (instead of siMETTL3), and seeing that it is only the late viral transcripts the ones that are affected in their splicing patterns? This point is specially important as it is the main finding claimed both in title and abstract, and it should be performed with the KO cells, rather than siRNAs, which can show variable depletion of the targeted gene during a time course experiment.

Minor Comments:

1) The authors mention in the Introduction that adenoviral RNAs are known to contain RNA modifications, based on previous literature. Which of them is it known to contain, apart from m6A? Is m6A the most frequent one? Can the authors elaborate more on this in the introduction?

2) Results in Fig 2B show the motifs predicted for cellular m6A mRNAs. What is the predicted motif in the viral RNA? Would it be possible to predict it, as done for the cellular RNAs, despite the broad peaks?

3) The authors show that both siRNA and METTL3 KO completely depletes METTL3 expression (Figure 2e and 2f). But it is largely unclear if adenovirus infections are done simultaneously with the siRNA

treatments, or later in time. If done later in time, the authors should also show that METTL3 expression is not recovered after several hours post-infection with the adenovirus (i.e. when the experiments will actually be collected), showing that the infection with adenovirus does not change/minimize the effect of the siRNA. The timings of siRNA treatments with regards to the viral infections should be clarified in the manuscript methods section.

4) Some statistical analyses and details related to them seemed to be missing throughout the manuscript. Below some examples:

- Results Page 5 line 135 - "There was a modest increase in levels of FTO and ALKBH5". Quantified how? Was this reproducible across different western blot gels using independent biological replicates? Did the authors perform replicates of the western blots? I couldn't find this information in Methods section.

- Results Page 5 line 143 "we observed that METTL3, METTL14, WTAP and YTHDC1 relocalized from their diffuse nuclear pattern into ring-like structures". Where there biological replicates? How was this relocalisation quantified?

- Page 6- line 151 "these nuclear proteins are concentrated at sites of viral RNA synthesis during infection". How was this quantified?

5) Figure 1. Why don't the timings (hpi) of the Western Blot protein expression (Figure 1A, time=12,24,48hpi) match those of immunofluorescence (Figure 1B, time=18hpi)? Was there a reason for this change in time post-infection?

6) Page 6 line 167- "We identified 19 peaks in the forward transcripts and 6 in the reverse transcripts" however MACS2-generated peak areas were very broad. Can the authors provide statistics on how the peak narrowness-broadness was in the viral sequences, relative to the cellular sequences? Also, perhaps the authors could propose some justification of why they observe this phenomena? Also, wouldn't the coupling to METTL3 KO cells reveal which of these "peaks" are METTL3-specific?

7) Direct RNA sequencing leads to many incomplete sequenced reads (i.e. truncated reads). How were these handled? Were they discarded from the analysis? Which were the criteria used to "keep" a read as corresponding to a specific isoform?

8) The authors show that DBP does not show a decrease upon siMETTL3 or siMETTL15 depletion, in contrast to other viral genes (Fig 5A). Could it be that the m6A marks in these gene are in fact not METTL3-dependent? Did the authors check whether potential m6A sites in this gene actually non-canonical METTL3 DRACH motifs?

9) Supplementary data that I was unable to find but would be good to provide:

- For meRIP-seq, I was unable to find the stats of how many reads were sequenced, mapping, how many

peaks, width of peaks, etc (I couldn't find any supplementary tables on this).

- For DRS, I was unable to find the stats of how many reads were sequenced, mapping, how many peaks, width of peaks, etc (I couldn't find any supplementary tables on this).

- For DRS, I was unable to find the list of reproducible and non-reproducible m6A sites, e.g. The tables generated with the bam-readcount software with the G-tests and bonferroni corrections (i.e. potential m6A sites) should be provided as supplementary material. Similarly, the tables generated with the bam-readcount software with the G-tests and bonferroni corrections (i.e. m6A sites), corresponding to those that the authors consider as "true" m6A sites, should be provided as supplementary material.

- Code for going from the output produced from bam-readcount output to final peaks should be made available in the form of a script to allow future researchers reproduce the work if needed

- Can the authors make materials available in Addgene? (eg. METTL3 KO cells, plasmids, etc)

We begin by thanking the editor and all the reviewers for the time they dedicated to the review of our manuscript. We believe that we have addressed all the major comments, which has resulted in a significantly improved paper. We have now ruled out contributions from m⁶A-mediated RNA decay by performing transcriptional shut-off assays during METTL3 knockdown. We also strengthened our observation of increased intron retention in the absence of METTL3 by using nanopore long-read sequencing of the viral transcripts. We demonstrated that loss of METTL3 correlates specifically with loss of gene expression of viral transcripts that contained long introns and many splice sites. Our direct RNA sequencing approach for m⁶A site discovery was statistically compared to the gold standard meRIP analysis, as well as validated through comparison against Nanocompore, an alternative dRNA sequencing analytical method for RNA modification detection. We are happy to report that both methodologies perform comparably, and we have now codified the pipeline (DRUMMER) which is now freely available on GitHub (<https://github.com/DepledgeLab/DRUMMER>). Finally, we expanded our mechanistic analysis of the contribution of the nuclear m⁶A reader YTHDC1 to explain the observed splicing defect. These improvements are reflected in substantial changes to three of the original figures and the addition of 23 new panels in supplemental figures. We have addressed all reviewer's points as detailed below.

REVIEWER COMMENTS:

Reviewer #1:

In this paper, Price et al. use cutting-edge techniques to catalog the m6A methylation events across the adenovirus transcriptome and they explore the functional importance of those methylation events. Given the emerging roles of m6A in gene expression in viruses and their hosts, these studies are timely and important for the field. Overall, this is a strong manuscript. The adenoviral m6A analysis in Figures 2-4 are particularly rigorous and clearly presented. Furthermore, the observation that adenovirus recruits the m6A machinery to viral replication centers is interesting. The authors further explore the effects of m6A on viral transcript accumulation and processing and propose that methylation of viral transcripts is required for late gene splicing. They present reasonable data supporting this conclusion, but their interpretations are currently too strong given the data presented.

In particular, I do not think that the authors have ruled out the contribution of m6A to RNA decay in their system. The only assay addressing RNA degradation is a calculated half-life based on nascent RNA and input levels. Given the importance of this conclusion, it is essential to complement this calculation with a more direct (e.g. transcription shut-off) assay.

Moreover, I am not sure whether the half-life calculation is applicable to adenovirus infection. I think these assays assume the total RNAs have reached steady-state. If so, that assumption is not valid during viral infection where late gene transcripts continue to accumulate over time throughout infection. For example, the Fiber RNA continues to increase over the course of infection from 24-30hpi (Crisostomo et al 2019). If this is driven by changing transcription levels, then the assumption of constant transcription does not hold (ref 80). On the other hand, if RNA accumulates simply due to decay rate being slower than a constant transcription rate, then the value assigned to input will be different if the samples were

taken at 24 hpi compared to 30 hpi. The resulting half-life determinations would reflect this difference, but this distinction has nothing to do with changes in decay rates. I'm not an expert in decay calculations, so my reasoning may be faulty. Intuitively, however, it seems to present an issue in the analysis. Either way, it is important to use an orthogonal approach because the downstream interpretations completely depend on the conclusion that decay is unaffected.

The reviewer's interpretation is correct, half-life calculations based on nascent transcription assume steady state mRNA levels¹. We attempted to correct for this by using the shortest possible time for a 4sU pulse that has been experimentally validated to provide meaningful data (10 minutes)². We have changed our text to indicate that this method only gives an approximation of RNA half-life (Line 350). In addition, we have also performed transcriptional shut-off assays using DRB, which validate that there are no significant changes in RNA half-life of early or late transcripts upon METTL3 knockdown (Supplemental Figure 6a).

1. The reporter assay (Fig 6h-i) is potentially powerful and should be expanded upon because this is the only data supporting a direct role for methylation as opposed to indirect consequences of cellular responses to METTL3 knockdown. Several assays should also be relatively straightforward to perform that would strengthen their conclusions:

a. For the reporter assay and the other spliced/unspliced assays (Fig 6d-f), they should show the levels of the unspliced and spliced RNAs in the supplemental. This is important because if changes in the ratios are driven solely by decreased spliced RNA, this result is equally consistent with a splicing defect or a decay defect.

We have separated the spliced and unspliced RNA data in Supplemental Figure 6. As can be seen, the levels of unspliced Fiber transcripts are slightly decreased in the presence of METTL3 knockdown (approximately two-fold), however the decrease in spliced Fiber transcripts was more affected by METTL3 knockdown (approximately four-fold). These data lead to the splicing efficiency decrease of approximately two-fold. The early gene E1A transcript was included as a negative control and was not similarly affected.

b. The siMETTL3 in the mutant construct decreases, even though it's not statistically significant. Therefore, it is difficult to discern whether it's not significant because there's no change in levels or because of experimental variability. Even the WT samples only decrease by ~25%, so the decrease shown may be closely mirroring this effect. Showing the individual values as dots on the graph would help interpretation of these data and more replicates may also clarify the interpretations.

For all data panels in Figure 6 and Supplemental Figure 6 we have included the individual data points to highlight our experimental *n* and data variability. We repeated the splicing construct transgene experiment four times, and we believe the increased variability stems from a single outlier. Based on our new data (Figure 7F) that m⁶A-dependent splicing efficiency is negatively correlated to intron length, we propose this explains why our splicing reporter transgene (intron length of only 1073 nucleotides) does not perform as robustly as the long intron Fiber gene during actual viral infection.

c. Examination of intronless WT/Mut reporters would go a long way in solidifying the conclusion that splicing drives the changes in RNA abundance.

Serendipitously, the splicing reporter plasmid we use (PsiCheck2) contains an unspliced Firefly luciferase transcript driven by an HSV-1 TK promoter downstream of the spliced Renilla luciferase driven by the SV40 promoter. We performed qRT-PCR for Firefly luciferase in wildtype and mutant splicing reporter samples, plus or minus METTL3 knockdown, and saw no significant changes in RNA abundance. This has been added as Supplemental Figure 6K, and reported in Line 402 (*“There was no change in the accumulation of an unspliced Firefly luciferase RNA expressed from the same plasmid upon METTL3 knockdown”*).

d. Conversely, a reporter with a strongly spliced m6A-independent intron would be interesting to test.

To the best of our knowledge, such a transcript that has been proven to be independent of m⁶A-mediated effects has not been reported.

e. They should perform the YTHDC1 knockdowns to the reporter to further substantiate their other mechanistic data.

We have repeated the splicing transgene assay plus or minus YTHDC1 knockdown and show similar results to the METTL3 knockdown. We believe this has significantly strengthened our claim of a direct role of m6A in YTHDC1-dependent splicing regulation and have included the data as Supplemental Figure 6j, and reported our findings in Line 399 (*“The same transgene splicing assay was performed following YTHDC1 knockdown (Figure S6j). A similar decrease in splicing efficiency of the wildtype reporter was observed, with no change in the m6A mutated reporter”*).

2. Fig 7. The fully processed full-length RNAs decrease in siMETTL3 samples, so the spliced junction reads will also go down regardless of the mechanism. If the decreased splice junctions go down and a concomitant increase in IR is observed, the case for splicing is much stronger. However, only a very small increase in IR is observed. One could reasonably argue that the unspliced IR transcripts are quickly degraded. Alternatively, one could argue that the mRNA is less stable in the siMETTL3 samples. Therefore, the result is ambiguous with respect to mechanism. In addition, browser shots comparing siCtrl to siMETTL3 would be nice to see to help readers visually inspect the reads on the introns/exons.

We agree with this reviewer suggestion and believe that the relatively small increase in intron retention (IR) detected is due to both the short half-life of unspliced transcripts in the nucleus³, as well as the fact that our sequencing libraries were selected for mature poly(A) containing transcripts. While our nanopore long-read sequencing suffers from the same limitation of poly(A) selection, we nevertheless looked for evidence of IR upon METTL3 knockdown and have included these data, as well as representative genome browser shots, as Supplemental Figure 7. Since nanopore reads start from the 3' end, we were additionally able to select for only the reads indicative of the cleavage events leading to L1-L5 transcripts. We calculated the average read depth of unspliced versus spliced regions for each of these transcripts. As can be seen, we now report increased IR for all the late transcripts upon either knockdown or knockout of METTL3. We have reported these findings in Line 440 (*“...we were able to*

uniquely bin all reads that were cleaved at any of the L1-L5 late gene polyadenylation sites and assay for the presence of intron retention upstream of the first splice acceptor (Figure S7a). Comparing the METTL3 KO to Wildtype cells or siMETTL3 to siCTRL data revealed evidence of increased intron retention/poor RNA processing in every late transcriptional unit...”), as well as strengthened our discussion of the relatively small increases in IR (Line 427, “While the increase in intron retention was not as dramatic as the decrease in spliced late transcripts, this can be explained by the relative instability of unspliced RNA transcripts within the nucleus, and the bias of the poly(A) selection employed for sequencing in capturing mature transcripts.”).

Reviewer #2:

In this work, Price et al. used direct RNA long-read sequencing to precisely map m6A sites on the complex transcriptome of Adenovirus. The authors show that transcripts of both early and late viral genes are modified, but report a differential effect of m6A, where reduction in m6A affects mostly late but not early viral gene expression. Mechanistically, the author show that m6A is important for proper splicing of late adenovirus genes. In general the experiments are well executed and the manuscript is well written.

Below are few suggestion that can help supporting the conclusions and improving the clarity of the manuscript

The most substantial improvement can be if the authors will use their long read and short read m6A sequencing data to strengthen the direct connection between the splicing defect and m6A modification. Is the effect of METTL3 depletion on late gene expression is correlated or restricted to modified transcripts? Are the genes that are changing contain m6A? can the authors show that other transcripts that are less affected (like L1-52K) are not (or less) modified? Is the defect in splicing related to length of the introns? Number of splice sites?

We thank this reviewer for a comment that has led us to do more analysis which has substantially strengthened the findings of our manuscript. Using a combination of our short-read and long-read data, we set out to ask what RNA features correlate with the loss of expression we see after METTL3 knockdown. Loss of gene expression was not correlated to presence of m⁶A sites, or even number of m⁶A sites, as shown in Figure 7d. We subsequently found that loss of gene expression in a METTL3-dependent manner was correlated both with number of splice sites (Figure 7e), as well as strongly correlated with the length of the largest intron in the gene body (Figure 7f). We report these findings in Line 446 (*“Utilizing the power of our long-read dRNA sequencing approach to map m⁶A sites to unique RNA isoforms, we asked if any specific features of these RNAs correlated with the loss of expression we saw after METTL3 knockdown...”*), and discuss the results in Line 472 (*“The benefit of m6A-mediated splicing efficiency was directly correlated to the intron length of the transcript, implicating a role for m6A in processing of long transcripts with many alternative splicing choices”*).

Specifically, in figure 7C the authors use the long reads to show there is a general reduction in late viral gene expression but are the long reads also support there are more non-spliced transcripts? Since this data should allow to directly connect the RNA-isoform to m6A sites, can the authors show that unspliced RNA contain less m6A signal than their spliced counterparts?

Unfortunately, the poly(A) selection we employed for nanopore dRNA sequencing biases us towards detecting mature transcripts. Furthermore, incompletely processed transcripts that we do detect are of low read-depth and heterogenous read length, which makes them sub-optimal for our m⁶A detection pipeline. As such we are not able to show a lack of m⁶A signal in unprocessed transcripts at this time.

However, we did attempt to use our long-read data to address our claim of Intron Retention or more non-spliced transcripts present after loss of METTL3. As seen with our response to Reviewer 1 comment #2, we were able to mine our existing long-read data for evidence of unspliced transcripts that proceed to the L1-L5 polyadenylation sites, and we show that there is increased IR for all of these species upon METTL3 knockdown or knockout (Supplemental Figure 7).

Another important point is whether the defect in Splicing is limited to the virus? If the effect also seen in other cellular genes (with many splicing acceptor sites?) is there a general correlation between m6A and splicing defect? Or any other feature that correlate with splicing defect and can give some insights about about the molecular mechanism.

While our nanopore sequencing data was of insufficient depth to fully characterize the human transcriptome, we have performed high-depth Illumina sequencing of RNA from infected and mock-infected cells plus or minus METTL3 knockdown. We detect 958 human genes significantly upregulated and 1298 genes downregulated upon METTL3 depletion (shown in Response Figure 1a). However, when we performed similar analyses to find correlations, we found no trends that correlate with presence or number of m⁶A sites (as determined by our meRIP), number of exons (essentially number of splice sites), nor the longest intron within the gene body (shown in Response Figure 1b-c). Furthermore, we mined our data for alternative splicing using the algorithm rMATS⁴. While we found 1201 cassette exons and 426 mutually exclusive exons that were differentially regulated upon METTL3 depletion (FDR < 0.05 and IncLevelDifference > 10%), none of these changes were correlated with the above RNA features. While the adenoviral late transcripts we used to discover the effect of m⁶A on splicing efficiency are generated

Response Figure 1. Changes in cellular transcripts upon METTL3 depletion. (a) Volcano plot showing gene expression changes of cellular genes upon knockdown of METTL3 in the absence of infection. Significance cutoffs of 4-fold change and FDR of 0.05. **(b)** Scatter plot of significantly changing cellular transcripts from **a** plotted by degree of fold change and number of annotated exons within the transcript. Presence (gold dot) or absence (grey dot) of m⁶A anywhere within the gene body determined by our meRIP-seq data in uninfected A549 cells. **(c)** Data plotted as in **b**, but with log2 fold change vs longest intron within a given transcript. No significant Spearman correlation was

from what is essentially one heavily alternatively spliced and polyadenylated gene, we believe that the complexity of genes across the human transcriptome is masking such a simple correlation with METTL3-dependence and RNA structural features. Since these data are outside the scope of our study we have decided not to include the Response Figure 1 in the main text, however if the editor or reviewers think it is necessary to strengthen our manuscript we will include it as a supplemental figure.

Minor points:

1. Fig. 6f: can the authors also show a decrease in the viral titer following knockdown of YTHDC1?

Similar to knockdown of METTL3, we now show that knockdown of YTHDC1 does not affect genome replication (early stage) but does affect infectious viral production (late stage). This decrease in viral titer (approximately ~two-fold) was significant but is less than that seen for knockdown of METTL3, mirroring the late protein immunoblot shown in Figure 6g. These data have been added to Supplemental Figure 6h-i.

2. Fig. 1b+c: in this figure, the authors show the localization of m⁶A writer and nuclear reader, as well as RNA Pol II, around the viral DNA replication sites (marked by DNA binding protein (DBP)). The text suggest these m⁶A machinery components are recruited to the replication sites, similarly to other cellular factors during viral infection. However, this localization of m⁶A components might be a result of their (known) association with RNA pol II and not of an active recruitment to these factors by the virus.

It is very possible that the m⁶A associated proteins are brought to viral replication centers as a consequence of RNA pol II transcription. We have added text and citations to include the known association of these factors with RNA pol II, and have modified our previous statements. See Line 131 (*"...by a writer complex composed minimally of METTL3, METTL14, and WTAP that associates directly with RNA Pol II"*) and Line 151 (*"...these nuclear proteins are concentrated at sites of viral RNA synthesis, and not actively excluded or mislocalized, during infection."*).

3. Fig. 2d – why were GAPDH and HPRT chosen as controls? The meRIP-seq data shows that they are not modified?

GAPDH and HPRT were chosen precisely because they are not modified by m⁶A. These data confirm that our m⁶A RIP is working as intended, and that there is no change in the background level of these unmodified transcripts upon siMETTL3 depletion.

4. Fig. 6c: the primers for E1A spliced isoform, as it appears in the illustration, might also amplify the un-spliced transcript (the intron is short enough). It would be better to use a fw primer which sits on the exon-exon junction or at least explain how this option was excluded

We were worried about the possibility that our splice-specific primer design might be able to capture the unspliced 252 nucleotide intron of E1A-s. However, upon performing melt-curve analysis of our qRT-PCR amplicons (shown here in Response Figure 2) we find that the splice-specific E1A-s primer pair

amplifies a single product with the expected melt curve.

5. Fig. 6a: What are the values in this graph normalized to? The siControl of E1A is not 100%.

The values in Panel 6a are supposed to be normalized to the average of the four siCTRL replicates set to 100%. Upon re-examination it appears that there was a rounding error in Excel caused by taking the average after the exponential conversion instead of before. This has been corrected.

6. Fig. S4e: what is the difference between the two controls in infected cells?

These two control lanes are from two separate wells both treated with siCTRL. It is merely there to test for biological variability and western blot variability.

7. Fig. 7b and c: please add significance asterisks to all bars, the current presentation is quite confusing.

Done.

Reviewer #3:

In this work, Price et al investigate the role of m6A RNA modifications in adenoviral transcripts. This is a timely and interesting topic, and the authors address it with a nice combination of tools and technologies. To address the role of m6A modifications during adenoviral infection, the authors performed transcriptome-wide mapping using MeRIP-seq and direct RNA sequencing (DRS), and in the case of DRS, combined with the use of METTL3 KO cells. The main findings of the authors are: 1) that METTL3 is the enzyme responsible for placing m6A RNA modifications in adenoviral RNAs; 2) adenoviral m6A can be identified using DRS at transcript level and single nucleotide resolution and 3) that m6A in

adenoviral RNAs is important for splicing of late adenoviral transcripts, but not for early ones. However, based on the data presented in the manuscript unfortunately would need additional analyses/experiments/clarifications to support the claims. With regards to claim #1, the authors indirectly show that METTL3 knockdown causes decrease in expression of 2 viral transcripts using MeRIP-qPCR (Figure 2d); however, this does not necessarily mean that the viral m⁶A is placed by METTL3. Claiming that METTL3 is the enzyme responsible for adenoviral methylation is a major finding of the manuscript, however they only show this indirectly using MeRIP-qPCR, and considering that this is one of the major findings of the work, it should be validated using an additional technique such as mass spectrometry or immunonorthern of viral transcripts when infecting METTL3 KO cells vs control cells. With regards to claim #2, the first method (MeRIP-Seq) led to broad peaks, and the data was not analyzed in the context of viral RNA. Therefore, all results rely on the DRS analysis. However, the authors should try to compare DRS to other well-established techniques such as MeRIP-Seq (i.e. to repeat their first experiment, but also sequencing the METTL3 KO strain, which wasn't done), or use miCLIP (which produces single nucleotide resolution) instead of MeRIP-Seq. Moreover, it would be recommendable to compare their DRS predicted sites to those predicted using other DRS m⁶A prediction software, such as Tombo, EpiNano or NanoCompore, and see if their predicted sites (and consequently their conclusions) hold. With regards to claim #3, the authors suggest that m⁶A only affects splicing of late transcripts, but it is unclear whether this effect might be only seen in late transcripts because the authors used for this experiment siRNA-METTL3, rather than the METTL3 KO cells. Could it be that the siRNA takes time to show its effect and thus, only late viral transcripts are affected? Overall, this work is an important contribution to the field, and tackles very relevant and interesting questions; however, additional experiments/analyses/clarifications are needed to support the main claims of the paper.

Major Comments

1) Which is the expected abundance of m⁶A based on previous works? Which is the observed abundance based on the current work? Are the results obtained direct RNA sequencing data in agreement with previous m⁶A abundance results, in terms of abundance of the m⁶A modification?

Unfortunately, none of the techniques performed by us or others on adenovirus RNA address the direct abundance of m⁶A within viral transcripts. In the original work of Sommer *et al*⁵ it was discovered that m⁶A was present at cap structures and internal sites within the first two thirds of viral mRNA, and they were able to calculate a relative estimate of 4 m⁶A molecules per viral transcript. In the work of Chen-Kiang *et al*⁶ it was determined that m⁶A was added to high molecular weight adenoviral RNA in the nucleus, and much of this mark remained upon export to the cytoplasm with an estimation of 1 in 400 (0.25%) of total adenovirus nucleotides being methylated. Both of these numbers fit well with what is known about total abundance of m⁶A with cellular mRNA (~0.1%^{7,8}), as well as the average number of m⁶A sites our DRS detected for viral messages (4.16 per transcript). This information has been added to the introduction in Line 71.

2) The authors mention that meRIP-Seq was not ideal because it produced broad peaks. Why was this? Did the authors try to predict motifs within their peaks? Could the authors try a different peak-calling algorithm? What was the size of the peak in cellular RNA transcripts compared to viral RNA transcripts?

Other works have shown that m6A RIP-Seq could be done on viral RNA, as the authors mention in their Introduction. Could the authors perhaps try miCLIP (instead of MeRIP-Seq) on METTL3 KO cells to avoid the broad peaks? It would be key to validate the predicted m6A sites (main claim of the paper) by showing a consistency between DRS predicted m6A and well-established orthogonal methods to DRS based on Illumina, such as MeRIP-Seq, miCLIP or MaZTER-Seq. On the other hand, the authors performed MeRIP-Seq in infected and uninfected cells, but not in METTL3-KO cells. Wouldn't that help identify which of the "broad peaks" are in fact non-specific, and identify METTL3-specific peaks? This is indeed the design the authors did for DRS, which succeeded.

We believe the width of the peaks is due to extremely complex and overlapping nature of adenovirus transcripts, with many bona fide m⁶A methylation sites appearing within close proximity. The overlapping nature of many viral transcripts leads to unique problems and limitations that are hard to overcome with any short-read based sequencing approach⁹. Peak width is controlled by the chemical fractionation of RNA before immunoprecipitation such that short reads should “pile up” in a peak over the modified residue. Indeed, this appears to be the case when comparing to cellular m⁶A peaks (which are found in gene bodies spaced out over many kilobases and often not overlapping) which had an average peak width of 285-337 nucleotides and led to the easy recapitulation of the known DRACN

motif using a *de novo* motif finder HOMER¹⁰. In the case of the virus, the compact nature of the genome and the many overlapping transcripts means that multiple m6A peaks will intersect, and these cannot be adequately deconvoluted by any peak-calling software known to us. We do not believe that viral infection played a role in increasing peak widths, since peaks on the cellular transcriptome from mock-infected or Ad5-infected conditions yielded similar peak sizes (shown in Response Figure 3), and both recapitulated the known DRACN motif (see Figure 2b). We have addressed these concerns in the text at Line 170 (“Multiple peak callers were used, but ultimately none were successful in deconvoluting the large peaks due to the nature of overlapping short reads from multiple distinct viral transcripts. MACS2

peaks were retained for downstream comparison due to the robust differential peak calling ability of this particular software with regards to other meRIP-seq datasets.”).

We attempted to use many different peak calling algorithms including exomePeak, MeTPeak, and MeTDiff before settling on MACS2 as being the most robust peak caller at handling meRIP data in spliced transcripts¹¹. Of these, exomePeak led to the broadest peaks while MACS2 led to the most narrow. HOMER did not yield any significant motifs when applied to the 25 meRIP regions defined on the viral transcriptome, likely due to the small number of regions and large search space. However, when unbiased motif analysis was applied to the seven nucleotide region surrounding the raw DRS derived m⁶A sites (Supplemental Figure 1e) we recapitulated the known METTL3-dependent DRAC motif.

While techniques such as miCLIP or MAZTER-seq can both provide single nucleotide resolution, they would not aid us in being able to assign individual marks to specific transcripts due to the short-read nature of the techniques and overlapping nature of most of the adenovirus transcripts. In particular, MAZTER-seq relies on a methylation sensitive RNA nuclease that cannot cut at methylated A*CA trinucleotide repeats which is only seen in ~16% of all m⁶A sites^{11,12}. In addition, meRIP-seq is designed to be used without the need for METTL3 removal by comparing the antibody-enriched pulldown to unenriched total RNA. For mRNA in particular, loss of METTL3 before enrichment has been shown to reduce peaks by as much as 98%, and finds very few false-positives (512/25,695 sites)^{11,13,14}. Of the remaining false positives, the vast majority were in non-polyadenylated rRNA assumed to be contaminations within the poly(A) selection. As such, we do not believe that the addition of other short-read based m⁶A mapping strategies will add any mechanistic insight to our story about the benefits of long-read direct RNA sequencing detection based methods.

We do believe it is important to show consistency between our nanopore DRS-derived m⁶A sites and the meRIP-seq we performed, as asked by the reviewer. To this end, our DRS technique had excellent concordance with meRIP with 9/25 meRIP-seq regions not represented in our DRS, and only 8/53 single nucleotide DRS sites not located within the meRIP-seq regions. These results are significantly higher than expected by chance given the size of the adenovirus genome (Fisher’s Exact Test, p-values 1.6662×10^{-5} and 6.6648×10^{-5} , respectively). We have added this analysis to Supplemental Figure 1f-g, and have addressed this in the text at Line 234 (*“The overlap between antibody-based meRIP predicted regions and our dRNA-based approach was significantly higher than expected by chance...”*).

3) A key claim from this work is that METTL3 is the methylase responsible for methylating adenoviral RNA. The authors show that METTL3 knockdown causes decrease in expression of viral transcripts (Figure 2d); however, to my understanding, this does not necessarily mean that the viral m⁶A is placed by METTL3. An example of this is shown by the authors in Fig.5A, where the authors show that siMETTL3 doesn't lead to decreased DBP, despite DPB contains m⁶A sites predicted by their DRS. Thus, the meRIP-qPCR data shown is insufficient to draw the conclusion that METTL3 is responsible for viral RNA modifications. Additional validation is needed in this specific point because it is one of the major points of the manuscript. One possibility would be to measure m⁶A levels of the viral RNAs using mass spectrometry or immunonortherns of viral-infected parent vs viral-infected METTL3 knockout strains.

The reviewer is correct in that the slight decrease of DBP mRNA in Figure 5 upon METTL3 knockdown (simple qRT-PCR of total RNA after infection) does not imply causality in METTL3 applying a m⁶A mark to that transcript. However, the assay performed in Figure 2d is meRIP-RT-qPCR, and involves transcript-specific qPCR performed on RNA enriched by anti-m⁶A antibody plus or minus METTL3 knockdown and normalized to the input levels of RNA and background binding of IgG control antibodies. This technique as performed is directly orthogonal to the immunonorthern suggested, and thus we believe that we can imply causality in METTL3 applying m⁶A marks to both early viral transcripts (DBP), late viral transcripts (MLP), and the positive control cellular MALAT1. The approximately two-fold decrease after METTL3 siRNA is exactly what is expected after meRIP-qPCR based on prior publications^{11,15}. We have clarified this point textually on Line 178 (*“These results were normalized for the amount of input RNA, and demonstrate that the amount of m6A-marked RNA available for immunoprecipitation were all reduced in a METTL3-dependent manner for the early viral transcript DBP and late viral RNAs generated from the Major Late Promoter (MLP), as well as the positive control cellular transcript MALAT1”*).

4) Regarding the DRS data, the authors collapsed NNACN sites and focus on the analysis of these sites for the rest of the paper - but what about the remaining sites that don't meet the motif? Were there other sites predicted as "different" using the G-test after bonf correction that did not meet this motif? What proportion of 5-mers that passed the test in fact met the NNACN motif? Could it be that perhaps motifs that do not fall into this motif are actually the most abundant? The authors show in Figure 3C the distance of sites to NNACN data, but that is already after all filtering and masking. I think would be good to determine whether other sites that are not NNACN might indeed be highly represented among the k-mers that are depleted in METTL3-KO vs WT cells, i.e. to do the analysis "blind to the motif", rather than "guided by the motif". The motif should appear as a consequence, if it is the true motif. Could it be that the DBP m6A sites for example (whose expression was unaffected by METTL3 KO), are actually METTL3-independent?

We apologize for any confusion this has caused during review, and we hope the graphical and textual changes we have made throughout the manuscript will help to make our approach more clear. In brief, our technique employs two bioinformatic manipulations after the calling of candidate sites based on G-test discovered error rates: collapsing/masking and shifting. When we filtered for all G-test significant nucleotides, we found many sites that cluster within less than 5 nucleotides of an adjacent G-test candidate (Supplemental Figure 1c). This is likely due to the technical fact that RNAs are actually read by the nanopore within windows of ~5 nucleotides, and thus a single modification can increase the error rate of the surrounding window¹⁶. Thus, our first step was to take all significant G-test candidates that fell within five nucleotides of each other, and collapse/mask them to a single candidate site based on the highest G-test score (Figure 3b). This site does not have to be an adenosine within an AC dinucleotide core, but we found that it often is (Figure 3c) with only 5/53 sites (9.4%) not directly centered on an AC dinucleotide (New Data: Supplemental Figure 1e). Before performing any additional manipulation we applied an unbiased motif finder to the seven nucleotides surrounding all collapsed/masked candidates and were able to recapitulate the known METTL3 DRACN motif (Supplemental Figure 1e). Finally, for all sites that were not directly centered on an AC dinucleotide, we asked if it were possible to shift the candidate site by fewer than 5 nucleotides (due to the nanopore 5

nucleotide window). If any masked site were not within 5 nucleotides of an AC it would have been left alone, but in practice all G-test called m⁶A candidates were within 5 nucleotides of an AC. As such the shifted motif (Figure 3e) is even cleaner, and only contains 3/53 sites that did not perfectly correspond with the DRAC motif, all of which were GUACU.

To summarize our modifications to the manuscript, we changed panels 3b and 3e, and added additional data in support of our masking strategy to Supplemental figure 1e. To clarify further we have added text to Line 216 (“...so we implemented an additional filtering strategy to collapse all clustered candidates to one candidate site”) and Line 228 (“We subsequently extracted the seven base sequence centered on a collapsed candidate m⁶A site and generated a sequence logo (Figure S1e) that closely matched the m⁶A DRACH logo that we confirmed for m⁶A modifications in the human transcriptome”) and Line 230 (“Since all m⁶A candidate sites mapped to within five nucleotides of nearby AC, subsequent shifting of the motif center to the closest AC revealed that only 13% of sites did not perfectly recapitulate the DRACH motif (Figure 3e). Of these non-canonical motifs, three were DRACG and the remaining four were GUACU”).

5) Regarding the nanopore DRS data analysis, the authors use a specific pipeline which employs mismatch errors followed by a G-test. While this is theoretically correct, it is unclear how this method would compare to others. It would be good if the authors would show how would their predicted m⁶A sites look like if they had used other methods such as Nanocompore, EpiNano or Tombo. Which of them are robustly predicted by at least 2 out of 4 algorithms, for example? Would the conclusions of the work hold if the "m⁶A sites" would be those that are predicted in common by at least 2 different algorithms?

We see the value in addressing our DRS RNA modification pipeline (DRUMMER) against others and have attempted to do so. Unfortunately, the alignment aspect of EpiNano is hardcoded into the algorithm and is not flexible enough to allow the isoform-level analysis we performed. Development of Tombo has been halted by Nanopore as of May 2019 due to bugs and lack of sensitivity. Comparison with NanoCompore, however, has proved fruitful. By comparing their pipeline and ours we have discovered very high levels of concordance among predicted m⁶A sites. This is now reported in Supplemental Figure 2. These new data are addressed in the text at Line 262 (“To validate our data further, we also identified putative m⁶A sites using NanoCompore which performs comparative interrogation of signal level direct RNA-Seq data (dwell time and current intensity) to predict modified nucleotides. Using the same datasets as inputs, we identified 204 putative m⁶A sites at the isoform level: 93 of these were also reported by DRUMMER in all four comparisons, while a further 69 were also reported by DRUMMER in 1-3 comparisons. This left just 41 sites predicted by NanoCompore that were not identified by DRUMMER compared to 42 sites predicted by DRUMMER but not NanoCompore...”). We believe that rigorous benchmarking of these two pipelines against each other will take significant time and is outside the scope of the current manuscript. We intend to perform an extensive comparison which will be presented to the community as a separate project, and therefore hope the editor and reviewers are satisfied with the additional data we have reported.

6) The authors show in Figure 7 that METTL3 knockdown globally dysregulates adenoviral late RNA processing but not early one. This result is very interesting, but to my understanding, was only conducted at one specific time point (t=24hpi) and using siRNA-METTL3, not METTL3 KO cells. Could

these results be reflecting the fact that siMETTL3 takes a time until it has an effect, only causing the effect in splicing in late viral transcripts, but not in early viral transcripts? How can we rule out the possibility that siRNA did not affect early transcripts because there was still sufficient METTL3 mRNAs and proteins when the early viral particles were produced? If this possibility cannot be ruled out, could then the authors perhaps confirm this by doing the experiment using METTL3 KO cells (instead of siMETTL3), and seeing that it is only the late viral transcripts that are affected in their splicing patterns? This point is especially important as it is the main finding claimed both in title and abstract, and it should be performed with the KO cells, rather than siRNAs, which can show variable depletion of the targeted gene during a time course experiment.

We apologize for any confusion in the depiction of our siRNA experiments. All knockdowns were performed 48 hours prior to infection, as diagrammed in the new Supplemental Figure 3a. In addition, we saw no further decrease in METTL3 expression at the RNA level (New data: Supplemental Figure 3b) or protein level (Figure 5a) in the 24 hours following infection. Additionally, we now show that the splicing defect we see in Fiber transcripts happens both early (18 hpi) and late (24 hpi) after infection following 48 hour knockdown, and the splicing efficiency of E1A is unchanged (Supplemental Figure 6f-g). Finally, we do see a METTL3-dependent splicing effect in Fiber with knockout cells (Figure 6e), whereas E1A is still unaffected in the METTL3 KO cells (New data: Supplemental Figure 6e). These changes have all been incorporated into the text on Lines 276, 372, and 374.

Minor Comments:

1) The authors mention in the Introduction that adenoviral RNAs are known to contain RNA modifications, based on previous literature. Which of them is it known to contain, apart from m⁶A? Is m⁶A the most frequent one? Can the authors elaborate more on this in the introduction?

This has been elaborated on in Line 71 (*“These two studies also predicted adenovirus RNAs to contain on average four m⁶A modifications per transcript, along with low levels of methyl-5-cytosine, for a total of 1/400 (0.25%) of modified viral nucleotides.”*).

2) Results in Fig 2B show the motifs predicted for cellular m⁶A mRNAs. What is the predicted motif in the viral RNA? Would it be possible to predict it, as done for the cellular RNAs, despite the broad peaks?

As mentioned in major point #2, it was not possible to use HOMER to predict m⁶A motifs in the viral transcripts using meRIP data. However, using DRS we have discovered the known METTL3 motif in our unbiased (pre-shift) data (Supplemental Figure 1e)

3) The authors show that both siRNA and METTL3 KO completely deplete METTL3 expression (Figure 2e and 2f). But it is largely unclear if adenovirus infections are done simultaneously with the siRNA treatments, or later in time. If done later in time, the authors should also show that METTL3 expression is not recovered after several hours post-infection with the adenovirus (i.e. when the experiments will actually be collected), showing that the infection with adenovirus does not change/minimize the effect of

the siRNA. The timings of siRNA treatments with regards to the viral infections should be clarified in the manuscript methods section.

Details of transfection experiments have been schematized in Supplemental Figure 3a.

4) Some statistical analyses and details related to them seemed to be missing throughout the manuscript. Below some examples:

- Results Page 5 line 135 - "There was a modest increase in levels of FTO and ALKBH5". Quantified how? Was this reproducible across different western blot gels using independent biological replicates? Did the authors perform replicates of the western blots? I couldn't find this information in Methods section.

This was not manually quantified, but was a reproducible trend seen across independent immunoblots of three biological replicates of the experiment performed in Figure 1a, as well as indirectly seen again in the siCTRL samples of the immunoblot in Supplemental Figure 5a.

- Results Page 5 line 143 "we observed that METTL3, METTL14, WTAP and YTHDC1 relocalized from their diffuse nuclear pattern into ring-like structures". Where there biological replicates? How was this relocalisation quantified?

Since the size, shape, and number of viral replication centers are not uniform, nor is there a meaningful analogous compartment in uninfected cells, quantification of proteins surrounding *de novo* replication centers is challenging. However, these ring-like structures were seen across three independent biological replicates.

- Page 6- line 151 "these nuclear proteins are concentrated at sites of viral RNA synthesis during infection". How was this quantified?

As mentioned above, quantification was not performed but these ring-like structures have been well established in the adenovirus field as containing viral RNA, nascent RNA (BrU labeling), and enrichment for RNA Pol II, hnRNP proteins, and splicing accessories¹⁷.

5) Figure 1. Why don't the timings (hpi) of the Western Blot protein expression (Figure 1A, time=12,24,48hpi) match those of immunofluorescence (Figure 1B, time=18hpi)? Was there a reason for this change in time post-infection?

The initial timecourse (12, 24, 48 hpi) was chosen to highlight events that happen very quickly, intermediately, or at late times during adenovirus infection. Later in the manuscript 18 hours was chosen since this time represents the major inflection point between the early and late stages of infection based on our previous work.

6) Page 6 line 167- "We identified 19 peaks in the forward transcripts and 6 in the reverse transcripts" however MACS2-generated peak areas were very broad. Can the authors provide statistics on how the peak narrowness-broadness was in the viral sequences, relative to the cellular sequences? Also, perhaps

the authors could propose some justification of why they observe this phenomena? Also, wouldnt the coupling to METTL3 KO cells reveal which of these "peaks" are METTL3-specific?

We have added data on the peak width for mock-infected cellular transcripts, Ad5-infected cellular transcripts, and viral transcripts as Response Figure 3. As discussed in major point #2, meRIP performed on mRNA appears very specific for METTL3-dependent m⁶A peaks, and others have shown that METTL3-KO or KD is not necessary.

7) Direct RNA sequencing leads to many incomplete sequenced reads (i.e. truncated reads). How were these handled? Were they discarded from the analysis? Which were the criteria used to "keep" a read as corresponding to a specific isoform?

For exome-level analysis all reads that were able to align to the Ad5 genome were used. For the transcript isoform-level analysis reads that could not be unambiguously mapped to a specific transcript, including truncated reads, were discarded. This was performed by requiring the reads that began at a designated 3' poly(A) tail to extend to within 50 nucleotides of the annotated TSS, or within the third exon of the late transcripts tripartite leader. This has been further clarified in the methods section on Line 753 (*"Unambiguous transcripts were defined as those that began at a designated 3' poly(A) tail and extended to within 50 nucleotides of an annotated TSS, or within the third exon of the late transcripts tripartite leader"*).

8) The authors show that DBP does not show a decrease upon siMETTL3 or siMETT15 depletion, in contrast to other viral genes (Fig 5A). Could it be that the m6A marks in these gene are in fact not METTL3-dependent? Did the authors check whether potential m6A sites in this gene actually non-canonical METTL3 DRACH motifs?

As mentioned in major point #3, the meRIP-RT-qPCR performed in Figure 2d verifies that the DBP transcript contains METTL3-dependent m⁶A marks, even if the knockdown of METTL3 does not lead to the decrease of DBP *total* RNA (Figure 5d) or protein (Figure 5a). We propose that while DBP is marked by m⁶A, its transcript structure is relatively simple with short introns (Figure 7), and does not require m⁶A for splicing efficiency. All m⁶A sites within the DBP transcript were of the canonical DRACN motif.

9) Supplementary data that I was unable to find but would be good to provide:

- For meRIP-seq, I was unable to find the stats of how many reads were sequenced, mapping, how many peaks, width of peaks, etc (I couldn't find any supplementary tables on this).

- For DRS, I was unable to find the stats of how many reads were sequenced, mapping, how many peaks, width of peaks, etc (I couldn't find any supplementary tables on this).

- For DRS, I was unable to find the list of reproducible and non-reproducible m6A sites, e.g. The tables generated with the bam-readcount software with the G-tests and bonferroni corrections (i.e. potential m6A sites) should be provided as supplementary material. Similarly, the tables generated with the bam-readcount software with the G-tests and bonferroni corrections (i.e. m6A sites), corresponding to those that the authors consider as "true" m6A sites, should be provided as supplementary material.

- Code for going from the output produced from bam-readcount output to final peaks should be made available in the form of a script to allow future researchers reproduce the work if needed

- Can the authors make materials available in Addgene? (eg. METTL3 KO cells, plasmids, etc)

All of these data have now been added to the source data excel spreadsheet that will be uploaded with publication. Additional data and reagents will be made available upon request to the authors.

References:

1. Russo, J., Heck, A. M., Wilusz, J. & Wilusz, C. J. Metabolic labeling and recovery of nascent RNA to accurately quantify mRNA stability. *Methods* **120**, 39–48 (2017).
2. Fuchs, G. *et al.* 4sUDRB-seq: measuring genomewide transcriptional elongation rates and initiation frequencies within cells. *Genome Biology* **15**, R69 (2014).
3. Doma, M. K. & Parker, R. RNA Quality Control in Eukaryotes. *Cell* **131**, 660–668 (2007).
4. Shen, S. *et al.* rMATS: Robust and flexible detection of differential alternative splicing from replicate RNA-Seq data. *PNAS* **111**, E5593–E5601 (2014).
5. Sommer, S. *et al.* The methylation of adenovirus-specific nuclear and cytoplasmic RNA. *Nucleic Acids Res* **3**, 749–65 (1976).
6. Chen-Kiang, S., Nevins, J. R. & Darnell, J. E. N-6-methyl-adenosine in adenovirus type 2 nuclear RNA is conserved in the formation of messenger RNA. *J Mol Biol* **135**, 733–52 (1979).
7. Meyer, K. D. *et al.* Comprehensive analysis of mRNA methylation reveals enrichment in 3' UTRs and near stop codons. *Cell* **149**, 1635–46 (2012).
8. Dominissini, D., Moshitch-Moshkovitz, S., Salmon-Divon, M., Amariglio, N. & Rechavi, G. Transcriptome-wide mapping of N(6)-methyladenosine by m(6)A-seq based on immunocapturing and massively parallel sequencing. *Nat Protoc* **8**, 176–89 (2013).
9. Depledge, D. P., Mohr, I. & Wilson, A. C. Going the Distance: Optimizing RNA-Seq Strategies for Transcriptomic Analysis of Complex Viral Genomes. *Journal of Virology* **93**, (2019).
10. Heinz, S. *et al.* Simple combinations of lineage-determining transcription factors prime cis-regulatory elements required for macrophage and B cell identities. *Mol. Cell* **38**, 576–589 (2010).
11. McIntyre, A. B. R. *et al.* Limits in the detection of m6A changes using MeRIP/m6A-seq. *Scientific Reports* **10**, 6590 (2020).
12. Garcia-Campos, M. A. *et al.* Deciphering the “m6A Code” via Antibody-Independent Quantitative Profiling. *Cell* **178**, 731-747.e16 (2019).
13. Schwartz, S. *et al.* Perturbation of m6A writers reveals two distinct classes of mRNA methylation at internal and 5' sites. *Cell Rep* **8**, 284–96 (2014).
14. Ke, S. *et al.* m(6)A mRNA modifications are deposited in nascent pre-mRNA and are not required for splicing but do specify cytoplasmic turnover. *Genes Dev* **31**, 990–1006 (2017).
15. Gokhale, N. S. *et al.* Altered m6A Modification of Specific Cellular Transcripts Affects Flaviviridae Infection. *Molecular Cell* (2019) doi:10.1016/j.molcel.2019.11.007.

16. Garalde, D. R. *et al.* Highly parallel direct RNA sequencing on an array of nanopores. *Nat Methods* **15**, 201–206 (2018).
17. Pombo, A., Ferreira, J., Bridge, E. & Carmo-Fonseca, M. Adenovirus replication and transcription sites are spatially separated in the nucleus of infected cells. *EMBO J* **13**, 5075–85 (1994).

REVIEWER COMMENTS

Reviewer #1 (Remarks to the Author):

With one exception, the authors have addressed my concerns. This is a strong manuscript with compelling data and mechanistic insights into the roles of m6A in adenovirus splicing.

The one exception is that their response to my concern regarding the equation used to estimate half lives is not sufficient. To be sure, taking a short transcription pulse time is appropriate and estimates are fine. However, the issue is that the RNA is not at steady-state, so the calculation is not applicable.

There may be some confusion regarding the definition of steady-state because the definition of steady-state in the referenced Russo et al. is not strictly correct. Russo et al. states that steady-state refers to no change in transcription, decay, accumulation over the course of the pulse (Fig 2 legend). However, steady state is when transcription rates equal decay rates resulting in no change in total RNA over time.

As a result, it's the TOTAL RNA levels that are problematic for their viral analysis. For example, let's say a viral RNA has a constant transcription rate and a constant decay rate of $t_{1/2}=10$ hrs. If the gene is induced at 20hpi the RNA will begin to accumulate over the course of infection. Even though the transcription and decay rates are the same, at 26 hpi there's more RNA than at 24 hpi which has more than 22 hpi. In this case RNA accumulates over time even though transcription and decay rates remain constant. Because the total amount changes their DR will change as will the calculated half-life. My concern is not that it's a $t_{1/2}$ estimate, but that the calculation is not valid under the experimental conditions. Therefore, it should be removed.

I strongly recommend taking the decay experiments from the supplemental data (Fig S6a) and placing them in the main text (Fig 6a/b) and replacing the current data. It's understood that transcription shut-off has its issues (toxicity), but the data are quite strong using this conventional assay.

Reviewer #2 (Remarks to the Author):

The authors have addressed all my concerns and have generated an impressive work that highlight the role of m6A in regulating splicing of a viral transcripts.

We begin by thanking the editor and the two reviewers for the time they dedicated to the review of our manuscript. After the second round of revision we believe we have satisfactorily addressed all reviewer comments and produced a superior manuscript. All changes to the presentation of source data have been made and incorporated into the body of the text as Tables 1-5, which will facilitate sharing of our data.

REVIEWER COMMENTS:

Reviewer #1:

With one exception, the authors have addressed my concerns. This is a strong manuscript with compelling data and mechanistic insights into the roles of m6A in adenovirus splicing.

The one exception is that their response to my concern regarding the equation used to estimate half lives is not sufficient. To be sure, taking a short transcription pulse time is appropriate and estimates are fine. However, the issue is that the RNA is not at steady-state, so the calculation is not applicable.

There may be some confusion regarding the definition of steady-state because the definition of steady-state in the referenced Russo et al. is not strictly correct. Russo et al. states that steady-state refers to no change in transcription, decay, accumulation over the course of the pulse (Fig 2 legend). However, steady state is when transcription rates equal decay rates resulting in no change in total RNA over time.

As a result, it's the TOTAL RNA levels that are problematic for their viral analysis. For example, let's say a viral RNA has a constant transcription rate and a constant decay rate of $t_{1/2}=10$ hrs. If the gene is induced at 20hpi the RNA will begin to accumulate over the course of infection. Even though the transcription and decay rates are the same, at 26 hpi there's more RNA than at 24 hpi which has more than 22 hpi. In this case RNA accumulates over time even though transcription and decay rates remain constant. Because the total amount changes their DR will change as will the calculated half-life. My concern is not that it's a $t_{1/2}$ estimate, but that the calculation is not valid under the experimental conditions. Therefore, it should be removed.

I strongly recommend taking the decay experiments from the supplemental data (Fig S6a) and placing them in the main text (Fig 6a/b) and replacing the current data. It's understood that transcription shut-off has its issues (toxicity), but the data are quite strong using this conventional assay.

Upon careful consideration of the original Russo *et al* paper and the Reviewer's comments, we agree that the half-life estimates achieved using the ratio of 4sU nascent RNA to total RNA utilize assumptions that do not hold true during viral infection with changing transcription rates. As such, we have entirely removed the data presented in Figure 6b and the accompanying methods, and replaced it with the transcriptional shut-off data that was present in Supplemental Figure 6a.

However, we would like to keep the 4sU-based *transcription* data that we presented as Figure 6a. This calculation is not based upon any ratios or assumptions, and simply asks the question how much of a given transcript was produced between two conditions in a given time frame. As such, the data is still crucial for us to make our interpretation that in the presence or absence of METTL3, similar levels of both early and late transcripts were transcribed in Ad5-infected cells.

Reviewer #2:

The authors have addressed all my concerns and have generated an impressive work that highlight the role of m6A in regulating splicing of a viral transcripts.

We are happy that our work was satisfactory to this Reviewer, and thank them for the effort the put into the initial review that substantially strengthened our manuscript.

Editor's comments:

1) Please address remaining concern of Reviewer #1 by removing fig 6a/b and move fig S6a to main figures.

Done.

2) Please provide below 4 tables in the source data file as supplementary data (in excel file format) and cite in the text.

-Illumina sequencing stats

-dRNA sequencing stats

-Exome-level DRS m6A analysis

-Isoform-level DRS m6A analysis

The Illumina sequencing stats have been transitioned to the included excel spreadsheet labeled Table 1. The dRNA sequencing stats are present as Table 3. Exome-level and isoform-level DRS m6A analysis are now included as separate tables Table 4 and Table 5, respectively.

3) Please provide a supplementary data file that includes number of peaks and width of peaks in meRIP-seq data.

The number, location, and peak width of all MACS2-derived meRIP-Seq peaks is now included in Table 2, and includes the raw peaks in human transcripts in Uninfected A549 cells, human transcripts in Ad5-infected A549 cells, Ad5 transcripts, and a summary of peak number and width between these conditions.